# Changes in the distribution of fitness effects and adaptive mutational spectra following a single first step towards adaptation

Dimitra Aggeli[1,3,5], Yuping Li [2,4,5] & Gavin Sherlock [1✉]

Historical contingency and diminishing returns epistasis have been typically studied for relatively divergent genotypes and/or over long evolutionary timescales. Here, we use *Saccharomyces cerevisiae* to study the extent of diminishing returns and the changes in the adaptive mutational spectra following a single first adaptive mutational step. We further evolve three clones that arose under identical conditions from a common ancestor. We follow their evolutionary dynamics by lineage tracking and determine adaptive outcomes using fitness assays and whole genome sequencing. We find that diminishing returns manifests as smaller fitness gains during the 2nd step of adaptation compared to the 1st step, mainly due to a compressed distribution of fitness effects. We also find that the beneficial mutational spectra for the 2nd adaptive step are contingent on the 1st step, as we see both shared and diverging adaptive strategies. Finally, we find that adaptive loss-of-function mutations, such as nonsense and frameshift mutations, are less common in the second step of adaptation than in the first step.

[1] Department of Genetics, Stanford University School of Medicine, Stanford, CA, USA. [2] Department of Biology, Stanford University, Stanford, CA, USA. [3] Present address: Department of Biological Sciences, Lehigh University, Bethlehem, PA, USA. [4] Present address: Department of Microbiology and Immunology, UCSF, San Francisco, CA, USA. [5] These authors contributed equally: Dimitra Aggeli, Yuping Li. ✉email: gsherloc@stanford.edu

Stephen Jay Gould argued that historical contingency makes evolutionary outcomes largely unpredictable, and that were we to replay the "tape of life", we would likely end up with a different world each time[1]. However, frequently observed instances of both parallel[2–4] and convergent[5–7] evolution suggest that, at least under some circumstances, adapting populations may simply take different paths to the same peak on a fitness landscape. Environmental similarities, genotypic relatedness, and proximity to an optimum in the fitness landscape are some of the constraints contributing to convergent or parallel adaptive responses[4,8–21].

Closely related genotypes are often employed to study the effects of evolutionary history on adaptation in various experimental systems[22–30]. A frequent observation is that fitness gains decrease over time during adaptive evolution—termed diminishing returns—most convincingly demonstrated in cases where founders with differing initial fitness are used[24,25,27,28,30–32]. However, support for the role of historical contingency during adaptation is not uniformly consistent. For example, evolutionary history has been shown to both contribute[23] and not contribute[26] to defining subsequent adaptive mutational spectra in closely related *Pseudomonas aeruginosa* lineages, while historical contingency in related evolving *Escherichia coli* populations manifested at a phenotypic but not at a molecular level[29]. By contrast, evidence of first-step adaptive mutations in *Saccharomyces cerevisiae* being mutually exclusive due to reciprocal sign epistasis[22,33,34] is clearly supportive of historical contingency. Nevertheless, experiments founded with related *S. cerevisiae* clones spanning a range of fitness effects, suggest that convergence at a molecular level can and does occur[30]. Such apparently contradictory results may stem from differences in evolutionary timescales, population sizes, or culture conditions. For example, longer timescales (up to several hundreds of generations) that allow for the rise of lineages to frequencies sufficient for easy detection via sequencing, also result in clonal interference, a consequence of clonal propagation in well-mixed environments[24,35–39]. Given sufficient time, clonal interference will result in a somewhat predictable outcome because competition among many adaptive lineages will reproducibly favor fixation of those with the highest fitness[30,36,40–42], even when there is genotypic divergence[30]. This suggests that long-term evolution experiments are limited in their ability to capture the full spectrum of adaptive mutations and their fitness effects. A prior example in *E. coli* has illustrated phenotypic diversity specifically among first step adaptive mutants, exemplifying the benefits of short-term experiments[43]. Additionally, timescales that allow for accumulation of several mutations are typically unable to resolve how early during adaptation historical contingency and diminishing returns manifest.

The application of molecular barcoding to experimental microbial evolution (EME), for the purpose of tracking lineages, has enabled high-resolution characterization of evolutionary processes[4,20,21,41,42,44], importantly, on shorter timescales (less than a few hundred generations). Such studies have revealed a plethora of available adaptive mutations that increase in frequency early in the evolution, but most of which will go extinct due to being outcompeted by high fitness lineages[4,41,44]. Here, we use DNA barcoding to investigate how closely related genotypes of *S. cerevisiae*, each with a single adaptive mutation relative to their common ancestor, further evolve in the environment under which they were originally selected[4,44]. The molecular barcodes allow us to detect lineages at very low frequencies, necessary to generate the distribution of fitness effects (DFE) during the 2nd step of adaptation. Barcoding informs our decision on the evolution timepoint(s) from which to isolate clones, so as to maximize representation of adapted lineages. In follow-up experiments, barcoding also allows for pooled fitness assays and whole-genome sequencing of independently arisen adaptive lineages. This allows us to deeply characterize the beneficial mutational spectra and fitness effects, and draw direct comparisons among second-step and between first and second-step adaptations.

The evolutionary environment we use is serial-transfer under glucose-limitation, where cells undergo lag, fermentation, and respiration phases within each growth cycle. Common adaptive strategies in the 1st step of adaptation in this environment include diploidization and upregulation of the Ras/PKA and TOR/Sch9 pathways[4]; our haploid founders for the 2nd-step evolutions carry either a *cyr1*, a *gpb2* (both of which upregulate the Ras/PKA pathway), or a *tor1* (which upregulates the TOR/Sch9 pathway) mutation. All three of these mutants have an increased cell size and a higher fitness relative to their ancestor[4,20], though their fitness advantages manifest differently within lag, fermentation, and respiration growth phases[20]. Diploidization is frequently observed during experimental evolution with haploid *Saccharomyces cerevisiae*[4,45–50], with estimated rate of $2 \times 10^{-5}$ events per cell division[45], while its fitness advantage depends on both the environment and the strain background[4,45–49]. Circumstantial evidence suggests that diploidization is still available as an adaptive strategy even following other adaptive changes. However, it is unclear whether diploidization remains prevalent after the first step of adaptation, and if it does, whether it abides by diminishing returns.

We evolve barcoded populations of each of these three mutants and characterize rates of adaptation, and the distributions of fitness effects (DFE) of second step mutations. We then isolate hundreds of independent adaptive lineages and perform whole-genome sequencing and fitness remeasurements. In all three of these different genotypes, we find that diploidization remains a prevalent adaptive strategy, occurring at a high rate and with a comparable fitness benefit in the adapted backgrounds as in their founder. In addition, we find that 2nd-step mutations confer a smaller fitness advantage than the 1st-step mutations in the respective backgrounds where they arise, consistent with diminishing returns epistasis. We also find that there is a partial overlap in the molecular basis of the 2nd-step of adaptation between genetic backgrounds: the TOR/Sch9 pathway mutant frequently adapts via mutations in the Ras/PKA pathway, while the Ras/PKA pathway mutants, *cyr1* and *gpb2*, sometimes acquire mutations in the TOR/Sch9 pathway. On the other hand, we rarely identify second-step mutations that further modify the same pathway. We also find that the spectrum of adaptive mutations shifts from affecting pathways that regulate the cell cycle and nutrient signaling to pathways that affect stress responses. Targets of selection include genes in the HOG, retrograde flow (RTG), and glutathione biosynthesis pathways. *GSH1*, which functions in the glutathione biosynthetic pathway, is a target of selection in all backgrounds, while the HOG pathway is targeted only in the TOR/Sch9 pathway mutant and the RTG pathway is mutated in the Ras/PKA mutants. The Ras/PKA pathway mutants have similar relative changes in fitness and adaptive mutational spectra to one another, that differ from those of the TOR/Sch9 pathway mutant. Finally, we find that the second step mutations are less likely to be disruptive (nonsense and frameshift mutations) compared to first step mutations. Altogether, our data show that a single adaptive change is sufficient to cause genetic divergence during the immediate subsequent adaptation, consistent with evolutionary history influencing future outcomes, and that the DFE between first and second step mutations differs, consistent with diminishing returns epistasis.

## Results

**Experimental design.** Previously, we evolved a population of barcoded haploid yeast cells in a 2-day serial transfer condition under glucose limitation and isolated thousands of evolved clones from cycle 11 (after ~88 generations)[4,44]. Such a timescale was long enough for adaptive clones to rise to a sufficient frequency in the population, while short enough that the majority of adaptive clones carries only a single causative mutation. We then measured the fitness of thousands of isolated clones under the evolutionary condition and whole-genome sequenced hundreds of adaptive clones to identify their causative mutations[4]. Two major adaptive strategies were observed: self-diploidization, and upregulation of nutrient-sensing pathways, including the Ras/PKA pathway and the TOR/Sch9 pathway[4]. We refer to this prior evolution experiment as the "1st-step evolution".

In this work, we chose three adapted clones from the 1st-step evolution. Compared to their common ancestor, each clone carries one of the following mutations: a presumptive gain-of-function (GOF) mutation in a positive regulator (*cyr1*) of the Ras/PKA pathway, a loss-of-function (LOF) mutation in a negative regulator (*gpb2*) of the same pathway, and a presumptive gain-of-function mutation in a positive regulator (*tor1*) of the Tor pathway (Table 1). GOF mutations were inferred by the regulatory role that the affected proteins have in the signaling pathway: we had previously observed that nonsense and frame-shift mutations frequently occur in negative regulators of Ras/PKA pathway (i.e., *gpb2*), while positive regulators of Ras/PKA (i.e., *cyr1*) and TOR/Sch9 (i.e., *tor1*) pathways solely had rare missense mutations[4]. Based on these data, we inferred that the missense mutations in these positive regulators were GOF. In this study, we used one mutant representing each of the following categories: a GOF (*cyr1*) and a LOF (*gpb2*) affecting the Ras/PKA pathway and a GOF (*tor1*) affecting the TOR/Sch9 pathway. No LOF mutant affecting the TOR/Sch9 pathway was identified in our prior work[4]. We intentionally chose these particular mutants to ensure that both GOF and LOF mutations were further evolved and that both signaling pathways were represented.

The founding populations were derived from adapted clones via backcrossing with a mating-type switched version of the unbarcoded wild-type ancestor (strain GSY5375, Table 1). Fitness advantages of the derived strains were validated and shown to be monogenic and segregate with the mutation (Supplementary Fig. 1, Supplementary Table 1). These derivatives were then re-barcoded and further evolved for 160 generations in the same environment. We refer to this further evolution as the "2nd-step evolution" (Fig. 1). We performed low coverage barcode sequencing (average of 27 reads per barcode for timepoint 0 and 12 reads per barcode for the rest; see Supplementary Data 1) of the populations over the course of the evolutions (Supplementary Fig. 2) and used these data to estimate the fraction of adapted individuals at each timepoint (Supplementary Fig. 3). Based on these data, as well as benomyl-based ploidy assays (Supplementary Fig. 4), we isolated thousands of clones from

cycles 20, 13, and 12, corresponding to generations 160, 104, and 96 (cells roughly divide 8 times during each cycle), from the "2nd-step evolution" of *cyr1*, *gpb2*, and *tor1*, respectively, where ~25–50% of the individuals in the population are estimated to be adaptive. Fitness remeasurements and genome-wide sequencing were conducted for these evolved clones isolated from the 2nd step evolution. Fitness estimates are expressed per generation (assuming 8 generations per growth cycle) for consistency with the bulk of the literature, although we are aware that fitness advantage is not equally distributed within the growth cycle[20]. In this study, we refer to the original ancestor used in the 1st-step evolution as the "wild-type" ancestor and we refer to the founders of the 2nd-step evolutions, *tor1*, *gpb2,* and *cyr1* mutants, as "adapted" ancestors.

**The distribution of fitness effects (DFE) is compressed for the second adaptive step.** We used lineage tracking data to estimate the distribution of fitness effects for each adapted ancestor from the 2nd-step evolutions and compared them to that of the wild-type ancestor from the 1st-step evolutions[44] (datasets 1 and 2 in ref. [41]) (Supplementary Fig. 2). Since the barcode sequencing depth was higher for the 1st-step evolutions (Supplementary Data 1), we down-sampled the 1st-step evolutions' data to a depth comparable to that of the 2nd-step evolutions and calculated fitness and fitness-dependent mutation rates (Supplementary Fig. 5). Fitness inference remained similar upon down-sampling (Supplementary Fig. 5A) and so did the mutation rate spectra for fitness above 4% (Supplementary Fig. 5B, C). This is likely a consequence of the faster adaptation of the wild type ancestor, resulting in very fit lineages dominating the population and neutral and lower fitness lineages thus being present at low frequency. By down-sampling, we essentially limited our ability to detect lineages below ~4% fitness per generation, largely represented by autodiploids. Despite the lower barcode sequencing coverage, lineages with fitness <0.04 in the 2nd-step evolutions were readily detectable, in contrast to the 1st-step evolutions at comparable coverage (Supplementary Fig. 5C). Thus, we used high coverage lineage tracking data from ref. [41] (datasets 1 and 2 without down-sampling) for the 1st step evolution, and lower coverage lineage tracking data for the 2nd step evolution to estimate the DFE.

Diminishing returns models of epistasis predict that the magnitude of fitness gains decreases as lineages approach a fitness optimum. Prior work has suggested that diminishing returns is at least partially due to decreased fitness gains as adaptive mutations accumulate within a lineage[24,25,27,28,30]; however, the DFE beyond the first step has not been characterized. Our data allowed us to directly compare the DFEs of two consecutive adaptation steps. We used the wild-type DFE of the first step and overlaid to the DFE of each of the adapted ancestors (Fig. 2, see Supplementary Fig. 6 for direct comparison of all DFEs). The density data were used to estimate the overall adaptive mutation rates as well as diploidization rates and the rate

**Table 1 Strains used in this study.**

| Strain | Genotype | Description | Reference |
|---|---|---|---|
| GSY5096/ SHA118 | *MATα ura3Δ* ybr209w::GalCre-natMX | Wild-type universal ancestor | [44] |
| GSY5375 | *MATa ura3Δ* ybr209w::GalCre-natMX | *MATa* version of the ancestor | This study |
| GSY6701 | *MATα ura3Δ* ybr209w::GalCre-natMX cyr1$^{S917Y}$ | Evolution founder | This study |
| GSY6702 | *MATα ura3Δ* ybr209w::GalCre-natMX gpb2$^{Y282*}$ | Evolution founder | This study |
| GSY6703 | *MATα ura3Δ* ybr209w::GalCre-natMX tor1$^{F1712L}$ | Evolution founder | This study |
| GSY5481 | *MATα ura3Δ* ybr209w::full barcode cyr1$^{S917Y}$ | Evolved clone | [4, 44] |
| GSY5128 | *MATα ura3Δ* ybr209w::full barcode gpb2$^{Y282*}$ | Evolved clone | [4, 44] |
| GSY5153 | *MATα ura3Δ* ybr209w::full barcode tor1$^{F1712L}$ | Evolved clone | [4, 44] |

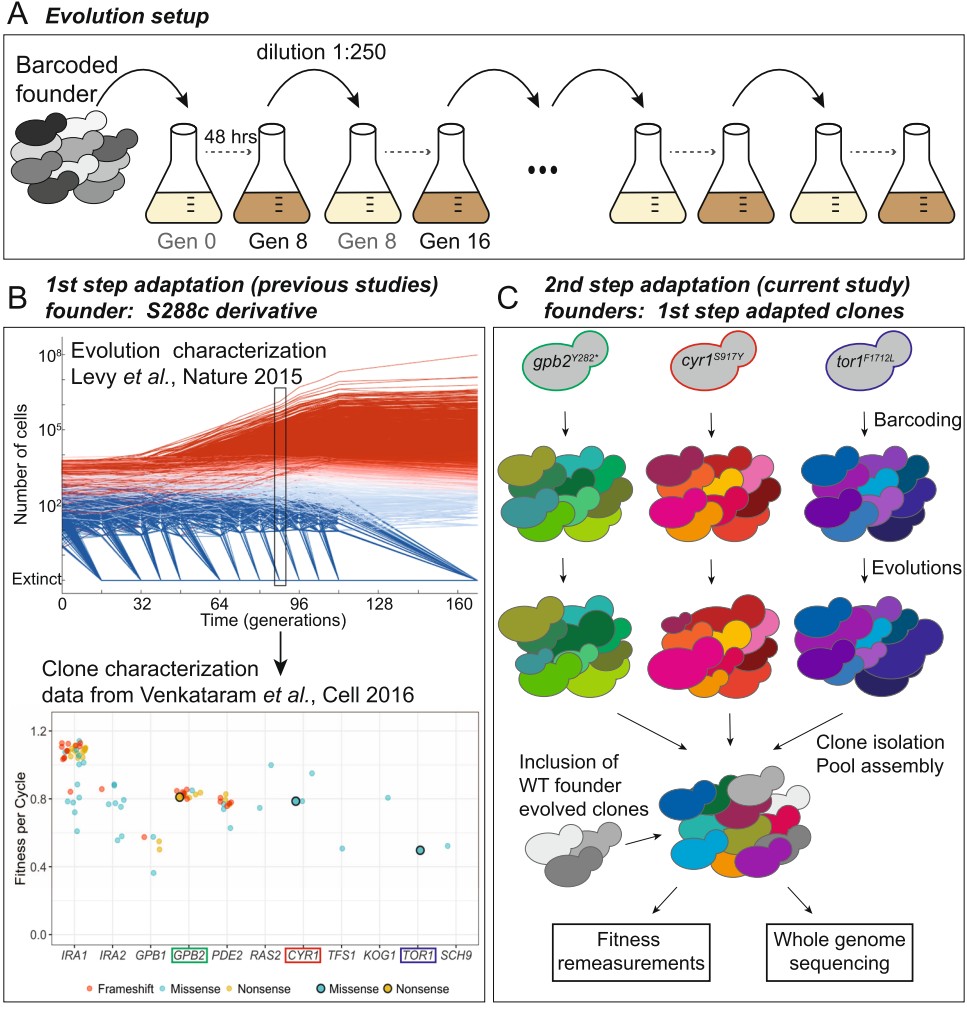

**Fig. 1 Experimental design. A** All evolutions were performed under identical conditions, including transfer and environmental conditions. **B** The founders used in this study derived from adaptive clones isolated and characterized previously[4,44]. The upper graph of (**B**) has been reproduced from ref. [44]. Red, cyan, and blue lines represent lineages with respectively diminishing probability of carrying an established beneficial mutation. The lower graph of (**B**) is a remake of Fig. 4 from ref. [4]. Different mutation types are color-coded as annotated. The ancestors of the evolutions of the current study are highlighted with an outer black circle. **C** Each founder carries a single adaptive mutation (gpb2[Y282*], cyr1[S917Y], and tor1[F1712C]), and was barcoded and propagated for 20 transfers. The mutations gpb2[Y282*], cyr1[S917Y], and tor1[F1712C] are color-coded green, red and blue, respectively. Different shades represent barcoded clonal derivatives. Gray represents the wild-type ancestor (annotated as WT). The same color scheme is used to annotate the ancestors or their evolved derivatives throughout the paper. Differences in yeast sizes represent barcode frequency changes. Adaptive clones are isolated and characterized via fitness re-measurements and whole-genome sequencing.

of occurrence of higher fitness mutations across different genetic backgrounds (Table 2). Note that the DFEs presented are truncated for lineages with s < 0.02, as this fitness coefficient approaches our lineage tracking data detection limit. While lineages with such small effect mutations might contribute to evolutionary outcomes of populations under weaker selection pressures (for example smaller bottlenecks or structured environments), they do not significantly affect outcomes in our experiments, which have large population sizes and are not mutation limited (see "Methods"). First, we observed similar shapes of DFE between the two evolutionary steps, featuring a long right tail and a peak at s = ~3–4% per generation across all genotypes. Our previous studies suggest that the peaks at 3–4% fitness likely correspond to lineages that have undergone autodiploidization, shown to be adaptive in the wild-type background[4]. Follow-up ploidy assays and fitness re-measurements of individual clones confirmed the prevalence and fitness advantage of diploidization during the 2nd-step evolutions (see below for details; importantly, the estimated

diploidization rate in the wild-type population is comparable to that previously reported[45]). We observed that 2nd-step evolutions manifest lower mutation rates over a wide fitness range beyond 0.04 per generation, and as the magnitude of fitness increases, the mutational fitness spectra decline at a faster rate compared to the 1st-step. Furthermore, adapted ancestors are devoid of very high fitness mutations compared to the wild-type ancestor, as demonstrated by the large difference in the mutation rates at fitness interval 0.07–0.12 and by the scarcity of lineages with fitness >0.12 for the adapted ancestors (Table 2). Finally, the beneficial mutation rates for the adapted ancestors were approximately one to two orders of magnitude lower than that of the wild-type ancestor (Table 2). In particular, the beneficial mutation rates for the wild-type ancestor were $1.57 \times 10^{-4}$ and $7.64 \times 10^{-5}$ for each of the 2 replicates, whereas for the adapted ancestors, the respective rates were $1.73 \times 10^{-6}$ and $2.61 \times 10^{-6}$ for cyr1, $9.68 \times 10^{-6}$ and $6.24 \times 10^{-6}$ for gpb2 and $9.17 \times 10^{-6}$ and $1.61 \times 10^{-5}$ for tor1. Overall, compared to the 1st-step adaptation, the 2nd-step adaptation with cyr1, gpb2, and tor1

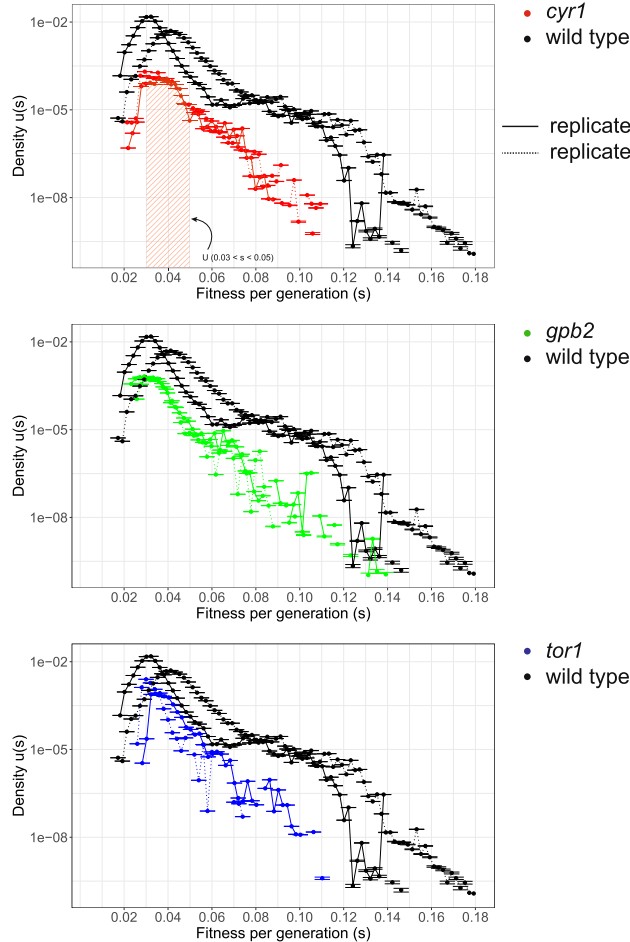

**Fig. 2 Mutation rates and fitness effects are smaller during the 2nd-step evolutions, as compared to the 1st-step.** Mutation rates per fitness bin were calculated for all 2nd-step evolutions and compared to the 1st-step (wild type). The integration of the area below the probability density curve represents mutation rate at the respective fitness interval. For instance, the shaded area in the top panel represents the mutation rate to generate mutants with a fitness value between 0.03 and 0.05. The complete wild-type evolution datasets were taken into account for the calculation (see Supplementary Fig. 5). The Y-axis error is defined in Eq. (2) and the error bars define the mutation rate ± the error (most error bars are sufficiently small so that the upper and lower handle appear to overlap). For a more direct comparison of all genotypes in a single plot see Supplementary Fig. 6. Mutation rates per fitness interval are provided in the Source data folder.

mutants not only have smaller magnitudes of fitness gains, as expected based on previous studies[25], but also have lower mutation rates for beneficial mutations within the detection limit, which has not been previously characterized. This suggests that diminishing returns in our system is driven by both declining fitness gains and decreased beneficial mutation rates. Based on this change in the DFE, we hypothesized that the adaptive genetic bases during the 2nd- step evolution will differ, opening up the possibility that they are contingent on the first adaptive step.

**Fitness gains of isolated adaptive clones from the 2nd-step evolutions tend to be smaller than in clones from 1st-step evolutions.** Having analyzed the DFE from the lineage tracking data during the evolutions themselves, we characterized the distributions of fitness effects of individual clones isolated from each of the 2nd-step evolutions. Clones were isolated from a single timepoint from each evolution and their fitness effects were

quantified against the wild-type ancestor under conditions identical to their evolutionary condition, by a bulk fitness assay. We directly compared the DFE between the 1st and the 2nd-step evolutions by including in our assays a set of isolated clones from the 1st-step evolutions[4]. Based on fitness and ploidy measurements, we classified isolated clones into four categories, consistent with the classification we used previously[4]. "Neutral haploids" are haploids with a similar fitness to their immediate ancestor, "adaptive haploids" are haploids with a higher fitness compared to their immediate ancestor, presumably carrying adaptive mutation(s), "pure diploids" are diploids without additional beneficial mutations, and "high-fitness diploids" are diploids with a fitness significantly higher than the mean diploid fitness, and likely harbor beneficial mutation(s) besides diploidy.

Fig. 3 shows the distributions of fitness effects per genotype, as calculated relative to the wild-type ancestor (Fig. 3A–D) and to their adapted ancestor (Fig. 3E–G). Isolated clones from the 1st-step evolution include the 2nd-step parental strains (corresponding points are annotated with larger dots in Fig. 3D), whose fitness value is included in Supplementary Table 1 (under "Fitness evolved remeasurements"). Deviations from earlier estimates[4] can be attributed to different population mean fitness resulting from inclusion of fitter strains in the pool. Compared to the wild-type ancestor, neutral haploids from the 2nd-step evolutions should have fitness comparable to their respective parental strains isolated from the 1st-step evolution. This is the case for the cyr1 and gpb2 genotypes (Fig. 3A, B), though neutral clones from the tor1 genotype evolution, whose fitness relative to wild-type is ~0.09 (Fig. 3C), have higher fitness than their unbarcoded ancestor, which had a fitness ~0.06 (represented with a blue dot in Fig. 3D), suggesting the possibility of the presence of mutation(s) that arose during the barcoding process. Overall, adapted clones from each of the three 2nd-step evolutions have further increased fitness compared to those from the 1st-step evolution (Fig. 3A–C). However, the fitness increase of this 2nd step is smaller than the fitness increase of the 1st step (Fig. 3D–G), suggesting a slower adaptation rate, consistent with the data from the lineage tracking during the evolutions. In particular, during the 1st-step evolution, adaptive clones gain benefits up to ~0.18 per generation compared to their WT ancestor. During the 2nd step evolutions, adaptive clones gain smaller fitness benefits compared to their immediate ancestors. The most fit clones gain benefits of ~0.09, ~0.10, and ~0.12 per generation compared to their cyr1, gpb2, and tor1 ancestors, respectively. Despite the small sample size, we observed an anti-correlation between ancestor fitness and highest fitness evolved (Supplementary Fig. 7, Pearson $r = -0.95$, Spearman $r = -0.8$ with $p$-values 0.049 and 0.333, respectively), consistent with diminishing returns. We cross-validated our fitness estimates from the lineage tracking data from the evolutions and from the bulk competition assays, by plotting the estimates against each other (Supplementary Fig. 8). Fitness values of lineages for which fitness was inferred from the evolution data approximately match the fitness values from the competition data. Discrepancies between the two datasets are expected to reflect cases where a single barcode represents more than a single genotype in the fitness inference from the lineage tracking data.

**Molecular targets of adaptation are contingent upon the founding genotype.** To study the genetic basis of adaptation on the different genetic backgrounds, we performed whole-genome sequencing on hundreds of adaptive clones isolated from the 2nd step evolution. The genetic basis of the 1st step evolution has been previously characterized[4]. Table 3 summarizes the molecular targets per founder and Fig. 4 shows their overlap with respect to

**Table 2 Mutation rates per genotype and diploidization fitness calculated from the DFE shown in Fig. 2.**

| Genotype, evolution | Diploidization fitness (1st peak at mutation spectrum, Fig. 2) | Diploidization rate at peak (for ds = 0.02, Fig. 2) | High fitness (> 0.05) mutation rate (Fig. 2) | High fitness (0.07–0.12) mutation rate (Fig. 2) | High fitness (> 0.12) mutation rate (Fig. 2) | Detectable fitness mutation rate (Fig. 2) | Lower bound for diploidization rate |
|---|---|---|---|---|---|---|---|
| WT, evo1 | 0.032 | 1.52E−04 | 1.42E−06 | 5.18E−07 | 8.07E−10 | 1.57E−04 | 7.18E−06 |
| WT, evo2 | 0.041 | 6.76E−05 | 9.22E−06 | 7.57E−07 | 2.27E−08 | 7.64E−05 | 1.79E−06 |
| cyr1, evo1 | 0.036 | 1.61E−06 | 7.74E−08 | 6.92E−09 | 0.00E+00 | 1.73E−06 | 1.22E−07 |
| cyr1, evo2 | 0.029 | 2.06E−06 | 9.25E−08 | 6.38E−09 | 8.59E−13 | 2.61E−06 | 2.01E−07 |
| gpb2, evo1 | 0.029 | 9.10E−06 | 1.27E−07 | 1.14E−08 | 5.59E−12 | 9.68E−06 | 1.04E−07 |
| gpb2, evo2 | 0.034 | 6.11E−06 | 6.41E−08 | 9.48E−09 | 7.69E−14 | 6.24E−06 | 4.22E−07 |
| tor1, evo1 | 0.036 | 8.71E−06 | 3.39E−07 | 8.69E−09 | 0.00E+00 | 9.17E−06 | 1.06E−06 |
| tor1, evo2 | 0.030 | 1.58E−05 | 6.23E−08 | 8.56E−10 | 0.00E+00 | 1.61E−05 | 1.88E−06 |

Diploidization rate from DFE was estimated from the sum of all the fitness intervals between (diploidization fitness −0.01) and (diploidization fitness +0.01). We chose the 0.02 fitness interval to match our detection limit. Mutation rates are expressed as the number of mutations per cell and generation. Fitness is expressed as fitness coefficient, estimated from the slope of the natural logarithm of adapted individuals over the neutral (ln(adapted/neutral)) over time.

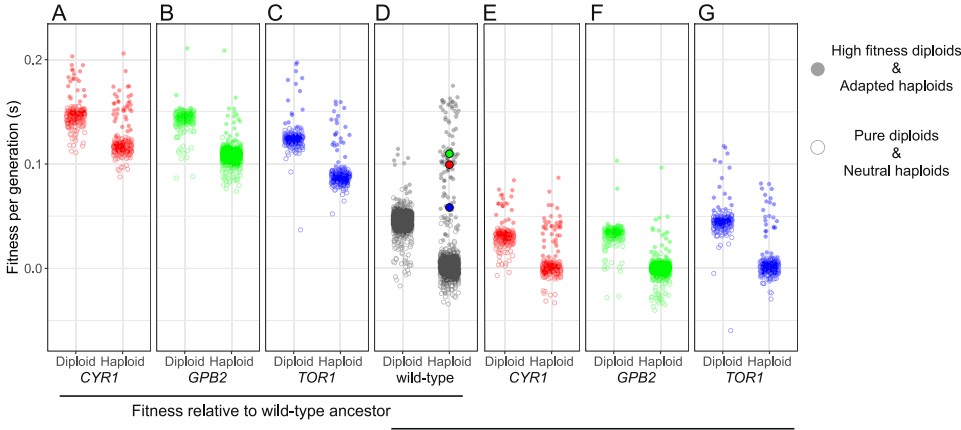

**Fig. 3 Distribution of fitness effects of 1st-step adapted clones following further adaptation.** Fitness values of isolated clones are shown with respect to the wild-type ancestor (A–D), and with respect to their immediate adapted ancestor (D–G). Fitness was measured in a pooled fashion from isolated clones of all immediate ancestor evolutions (including the wild-type), and arranged by ancestor (as annotated at the bottom). Haploids and diploids are shown in separate columns for each genotype. Clones with increased fitness within each group (high fitness diploids and adapted haploids) carry different annotation from pure diploids and neutral haploids. Larger dots in the ancestor cloud represent the adapted ancestors prior to barcoding and are color-coded to match the respective genotypes. Fitness values are provided in the Source data folder.

genes and pathways that were mutated. We observed similarities and differences in the mutational targets between the 1st- and 2nd-step evolutions and among the 2nd-step evolutions.

Genes in the Ras/PKA and TOR/Sch9 pathways are the major adaptive targets during the 1st-step evolution and are also targets during the 2nd-step evolution. However, the *tor1* mutant is more likely to acquire adaptive mutations in the Ras/PKA pathway (6 out of 21 multi-hits in *tor1*, 1 out of 19 in *cyr1*, 0 out of 5 in *gpb2*), while *cyr1* and *gpb2* mutants are more likely to acquire mutations in the TOR/Sch9 pathway (1 out of 19 in *cyr1*, 2 out of 5 in *gpb2*, 0 out of 21 in *tor1*). The observation that double mutants on the Ras/PKA and TOR/Sch9 pathways are more fit than their corresponding single mutants and were selected for, whereas double mutants on the same pathway were not, suggests that the TOR and Ras/PKA pathways are not redundant in how they increase fitness, as has been previously shown[20,51].

In contrast to the 1st-step targets of selection, stress response pathways were major targets of selection during the 2nd evolutionary step. *GSH1* mutations were observed 8 times in total across all three genotypes of the 2nd-step evolution, yet no *GSH1* mutations were observed during the 1st-step evolution. Similarly, mutations affecting the retrograde (RTG) pathway were exclusively observed in the Ras/PKA mutant backgrounds (7 out

of 19 in *cyr1*, 1 out of 5 in *gpb2*, 0 out of 21 in *tor1*), while HOG pathway mutants were observed predominantly in the *tor1* mutant background (13 out of 21 in *tor1* including pre-existing mutations, 2 out of 19 in *cyr1*, 1 out of 5 in *gpb2*) and *aro80* was also only observed in the *tor1* mutant background (Table 3, Fig. 4).

Finally, the predominant adaptive mutation type differs between 1st- and 2nd-step evolution, in terms of the consequences the mutations have on the encoded protein. Adaptation via LOF mutations is common during the 1st-step evolution, whereas the 2nd-step of adaptation often selects presumptive GOF mutations. Adaptive changes that increase signaling in Ras/PKA and TOR/Sch9 pathways can be achieved either by LOF mutations in negative regulators, or, rarely, by presumptive GOF in positive regulators. Specifically, 53 out of 95 causative mutations (56%) from the 1st step evolution result in either a frameshift or stop-codon gain (nonsense), likely leading to the loss of function of the mutated gene. By contrast, only 14 out of 55 causative mutations (25%) from the 2nd step evolution are frameshift or stop-codon gain mutations. These calculations include both the common targets of selection in Table 3 and additional mutations that occurred on the background of a stronger causal mutation candidate (not included in Table 3, see

**Table 3 Genetic basis of adaptation per founder.**

| Pathway | Mutated gene | WT | cyr1 | gpb2 | tor1 |
|---|---|---|---|---|---|
| RAS/PKA | CYR1 | 3 | | | |
| RAS/PKA | GPB1 | 5 | | | |
| RAS/PKA | GPB2 | 13 | | | 2 |
| RAS/PKA | GPR1 | | 1 | | |
| RAS/PKA | IRA1 | 39 | | | 1 |
| RAS/PKA | IRA2 | 10 | | | 1 |
| RAS/PKA | PDE2 | 11 | | | 1 |
| RAS/PKA | RAS2 | 2 | | | 1 |
| RAS/PKA | TFS1 | 1 | | | |
| RAS/PKA | YAK1 | 1 | | | |
| TOR/Sch9 | KOG1 | 1 | | | |
| TOR/Sch9 | KSP1 | 1 | 1 | 1 | |
| TOR/Sch9 | MDS3 | 1 | | | |
| TOR/Sch9 | SCH9 | 2 | | 1 | |
| TOR/Sch9 | TOR1 | 1 | | | |
| HOG | HOG1 | | 1 | | |
| HOG | PBS2 | | | | 6 |
| HOG | SSK2 | 2 | 1 | 1 | 7 |
| RTG | BMH1 | | 1 | 1 | |
| RTG | MKS1 | | 2 | | |
| RTG | RTG2 | | 4 | | |
| RTG | GSH1 | | 5 | 1 | 2 |
| RTG | ARO80 | | 3 | | |

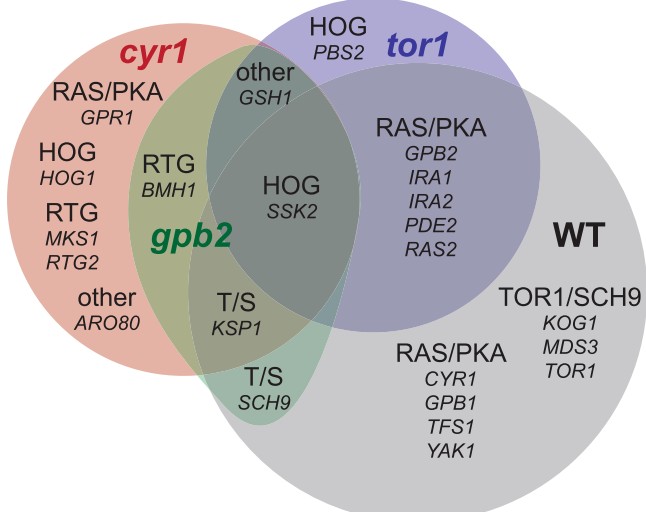

**Fig. 4 Overlap in the mutational recurrent targets among adaptive and wild-type ancestors.** The pathway and the gene names are annotated in each area. For number of hits, see Table 3. T/S stands for TOR/Sch9. WT stands for wild-type.

Source data for Fig. 3 for a more complete list). The beneficial mutation types between the 1st- and 2nd-step evolutions are significantly different (chi-square *p*-value = 6E−4), which also indicates a shift of beneficial mutation spectra during adaptation. The fact that the 2nd-step adaptation is more likely to result from GOF mutations may be partly responsible for the decreased mutational target size during the 2nd-step evolution, given that LOF mutations occur more easily than GOF mutations[52].

**Effects in pairwise targets of selection may be additive or negative**. Diminished fitness gains could result from beneficial alleles that show negative epistasis, and/or may reflect the order in which adaptive mutations are selected (higher fitness

mutations should be favored first). We estimated the extent of negative epistasis at the gene level. Availability of mutations affecting the same genes in both 1st- and 2nd-step evolutions, and of fitness effects of the implicated genotypes (single or double mutants), allows for a crude estimation of epistatic interactions among targets of selection. We considered the average fitness effects of alleles in two genes when they occur either singly (fitness averaged from all alleles of a gene, data from 1st-step evolutions) or together (fitness averaged from all genotypes that had the 2 genes mutated, data from 2nd-step evolutions) in the wild-type background, for genes where such data were available (Fig. 5). In all cases, 2nd-step adapted mutants are more fit than either of the 1st-step adapted mutants. The range of expected fitness for a genotype with both genes mutated, without epistasis between them (gray bar in Fig. 5), is represented by the 95% confidence interval (CI) of the sum of the mean fitness of mutants in each gene. Only *ksp1* in combination with either *cyr1* or *gpb2* was consistent with negative epistasis. The remaining combinations have fitness effects that do not deviate from the expectation of an additive model of epistasis (of log(fitness)). Thus, these data only provide weak evidence for the hypothesis that negatively interacting alleles contribute to diminishing returns in our experiments. However, we note that this analysis relies on alleles that were selected: not only did they emerge as adaptive, but they had a sufficiently positive fitness effect to gain in frequency and ultimately be picked. The allelic combinations may thus represent those with the least negative interactions. Overall, although the particular calculations do not provide support for negative interactions as a source of diminishing returns, negative interactions between alleles not included here may have contributed to a narrower DFE.

**Diploidization is adaptive across genotypes**. Diploidization is a major adaptation strategy during evolution experiments founded with haploid yeast[4,45–50]. During evolution of the wild-type ancestor, between ~32 and ~54% of the population was diploid by generation 88; the majority of these diploids does not carry additional adaptive mutations and have a similar fitness advantage to one another over their wild-type ancestor (~0.045 per generation)[4]. Diploidization remained an adaptive strategy after acquisition of a first adaptive mutation, as demonstrated by all three genetic backgrounds. We performed benomyl sensitivity assays to estimate the fraction of diploids during the evolution experiments (Supplementary Fig. 4A) and observed an increase in the fraction of diploid individuals over time (Supplementary Fig. 4B, Supplementary Table 2). At generation 88, the diploid fractions for these adapted mutants were on average 16%, 10 and 45% for the *cyr1, gpb2,* and *tor1* evolutions respectively, based on assaying ~60–190 individuals. Interestingly, diploidization in the *tor1* background approached fixation in both replicates (96 and 88%) at generation 160, unlike in the evolutions in the other mutant backgrounds. That may be due to chance or may instead reflect either an otherwise comparatively weak adaptation potential for the *tor1* mutant, or an increased fitness for diploidy and/or diploidization rate in a *tor1* background compared to the other backgrounds. In the Supplementary Note, we outline how different factors (i.e., diploidization rate, fitness of diploids, and the population mean fitness) quantitatively determine the frequency of diploids in an evolving population.

We used the fitness remeasurement data to calculate the fitness advantage of diploids in the context of different genetic backgrounds, including the wild-type ancestor. We observed two dominant groups of clones with distinct fitness (Fig. 3), which correspond to neutral haploids and pure diploids[4]. Pure

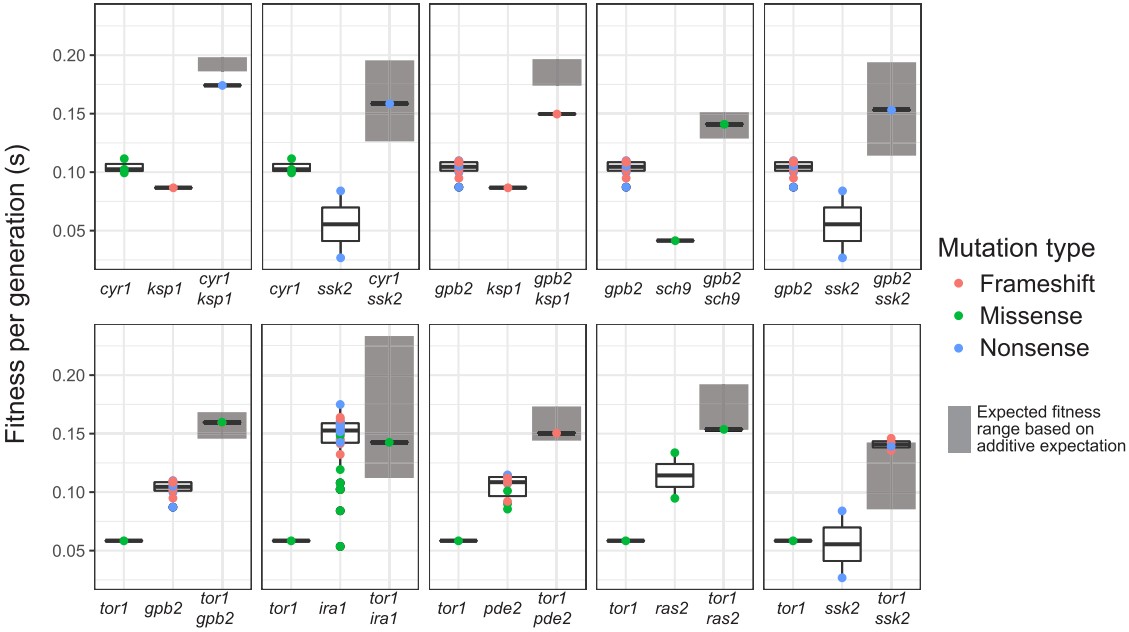

**Fig. 5 Pairwise interactions among targets of selection have additive or negative effects when combined.** The additive fitness effects were estimated at gene-level for pairs of genes with adaptive alleles on the wild-type background (data from 1st-step evolution), and are annotated with gray bars, representing the 95% CI of the sum of the mean fitness of mutants in each gene. The lower and upper hinges of each box correspond to the first and third quartiles (the 25th and 75th percentiles). The bold line inside the rectangular defines the median. The whiskers extend from the hinge to a value no further than 1.5 * IQR from the hinge (where IQR is the inter-quartile range). Additive fitness effects are compared to the fitness effects of genotypes with adaptive alleles at both genes (data from 2nd-step evolutions), as a proxy for epistasis. The dot color of the double mutants refers to the second mutation shown in each graph. The first mutation matches the mutation type of the adapted ancestors (*cyr1* and *tor1* are missense and *gpb2* is a nonsense). Calculated fitness values are provided in the Source data folder.

diploids from wild-type, *cyr1*, *gpb2* and *tor1* genetic backgrounds have fitness advantages of ~0.045 (95% CI [0.044, 0.045]), 0.028 (95% CI [0.027, 0.029]), 0.031 (95% CI [0.030, 0.032]) and 0.043 (95% CI [0.041, 0.044]) per generation, respectively, relative to their immediate ancestors (Fig. 3). Although, these values are similar, there is an anti-correlation between ancestor fitness and diploidization fitness advantage (Supplementary Fig. 7, Pearson $r = -0.91$, Spearman $r = -0.8$, with p-values 0.091 and 0.333, respectively), consistent with the observation that the highest fitness gains occur in the lowest fitness background. We also used the fitness spectra, shown in Fig. 2, from the lineage tracking data to estimate diploidization fitness advantage (Table 2). Diploidization fitness values were assumed to correspond to the peak within the fitness intervals ~3–4% per generation (Fig. 2; Table 2). Diploidization fitness across genotypes is comparable with the fitness estimates from the remeasurements data (Supplementary Fig. 7). We consider the values from the fitness remeasurements more accurate and comparable to each other, since clones were competing in the same pool against a mostly wild-type population.

To estimate a lower bound for diploidization rates, we used a second, orthogonal, and conservative approach (see Methods), independent from our estimates inferred from the DFE (Table 2). Taking into consideration only lineages whose fitness fall into the 95% CI of the experimentally verified diploids, the estimated lower bounds for diploidization rates per genotype and replicate evolution are 1.22e−07 and 2.01e−07 for *cyr1*, 1.04e−07, and 4.22e−07 for *gpb2*, 1.06e−06 and 1.88e−06 for *tor1* and 7.18e−06, and 1.79e−06 for wild-type, replicates 1 and 2, respectively (see "Methods"). Importantly, the lower bound and the diploidization rate estimates are correlated (Pearson Coefficient 0.99; *p*-value < 0.0001) and suggest that diploidization rates in the Ras/PKA mutant backgrounds are lower than those of wild-type

and the *tor1* mutant. These data collectively show that diploidization remains a prevalent adaptation strategy after acquisition of a first adaptive mutation.

## Discussion

We characterized the DFE and adaptive mutational spectra of three 2nd-step evolutions and compared them to a previously described 1st-step evolution[4]. We found that even a single step towards adaptation suffices to alter subsequent adaptation rates and adaptive mutational spectra. Use of molecular barcodes allowed us to characterize a large number of independent adaptive events and detect lineages at frequencies as low as 0.0001%, resulting in the most comprehensive characterization of DFE during the 2nd step of adaptation. The inferred DFE provides important data for the prediction of evolutionary outcomes.

Diminishing returns epistasis is apparent at multiple levels in our study, including the magnitude of fitness gains and the rate of adaptation. Both maximum and diploidization fitness anti-correlate with founder fitness (Supplementary Fig. 7), in agreement with diminishing returns acting globally[25,30]. We also observed a lower mutation rate to modestly adaptive genotypes, as well as a depletion of high fitness events compared to the wild-type ancestor. These diminished adaptation rates are also supported by fitness remeasurements of individual clones from the 2nd-step evolution. Overall, our observations imply that both a smaller number of adaptive mutations and smaller fitness effects cause diminishing returns. Whole-genome sequencing of adapted clones suggested that compared to the 1st step of adaptation, the 2nd-step was more often the result of presumptive GOF mutations. Such mutations are rarer[52], and that may provide a potential explanation for why we observe a smaller adaptive target in our experiments.

Comparison of adaptive strategies among the adaptive ancestors and between adaptive and wild-type ancestors reveals both common and distinct evolutionary paths. First, diploidization was a common adaptive strategy among all genotypes. Second, modification of both Ras/PKA and TOR pathways is another prevalent adaptation strategy, with the wild-type ancestor acquiring beneficial mutations in both Ras/PKA and TOR pathways during the 1st-step evolution and Ras/PKA and TOR adaptive ancestors further adapting during the 2nd step through modifications of the TOR and Ras/PKA pathways, respectively. We also observed (with a single exception) that the same pathway was not mutated a second time. These above observations suggest firstly, that the TOR/Sch9 and Ras/PKA pathways are not functionally redundant, consistent with earlier work[51], and secondly, that the highest fitness lineages are selected at each step due to the large and well-mixed populations used in this study. As a result, it is possible that in our isolated lineages the activity of the affected pathway is close to optimal under the evolutionary condition, and thus, second step variants further affecting the same pathway may move away from, or even overshoot such an optimum level, making them unlikely to be selected[53]. This is also consistent with prior observations where combining adaptive variants affecting the same pathway resulted in reciprocal sign epistasis in the environment in which they were selected[22,33]. Finally, common adaptive responses during the 2nd-step evolution included mutations that potentially result in the upregulation of stress response pathways, which we did not observe in the 1st-step. Both TOR/Sch9 and Ras/PKA pathways, which were major targets during the 1st step adaptation, regulate growth and stress responses responding to extracellular stimuli[51,54–57]. 1st-step adapted mutants had either the Ras/PKA or the TOR/Sch9 pathway upregulated[4,20], which may decrease stress responses compared to their ancestor[56–66]. The adaptive basis of 2nd-step mutations may lie in the restoration of stress responses attenuated by overactive Ras/PKA or TOR/Sch9 pathways, as suggested by prior work[51,57–66]. In particular, all adapted ancestors evolved in this study acquired mutations in the GSH1 gene, and the fact that all recovered GSH1 mutations were missense suggests that these mutations are GOF, possibly resulting in an enhanced stress response. Gsh1p catalyzes the first and rate-limiting step of glutathione biosynthesis[67], while no mutations affecting the pathway downstream were detected. We note that in another study glutathione export (both in oxidized and reduced forms) emerged as an adaptive response in cells experiencing nutrient–growth dysregulation, as a detoxifying mechanism[68]; it is unclear whether there is a specific connection between the adaptation observed in those experiments and that observed in ours.

We also observed specific adaptive routes available to either the tor1 mutant ancestor or the Ras/PKA mutant ancestors. Modifications of the retrograde pathway, including mutations in one positive (RTG2) and two negative (BMH1 and MKS1) regulators[69], were specific to the Ras/PKA mutants. Similar to our observations on the glutathione biosynthesis pathway, all 4 mutations on RTG2 were missense, while the 2 mutations on BMH1 included a missense and a nonsense and the 2 mutations on MKS1 included a missense and an upstream modification, suggesting that selection favors an enhanced retrograde flow pathway. Retrograde flow is negatively regulated by the TOR pathway[59,65,66], providing a potential explanation as to why modifications of this particular pathway were not observed in tor1 lineages. Modifications of retrograde flow, given an overactive TOR pathway, would need to be of larger effect to overcome the additional TOR-induced inhibition. Such large effect modifications may be rare, while tor1 lineages are able to improve via different, more easily accessible routes, such as via modifications of the HOG pathway. Despite the common overarching adaptation via modification of stress response pathways, specific mutational targets differ between tor1 and Ras/PKA adapted ancestors. Considering the overall adaptation patterns for all genotypes, it appears that a detectable fraction of the mutational spectra is at least partially contingent on the starting genotype, while phenotypic relatedness dictates the degree of divergence, as also shown by earlier studies[9]. We did not observe adaptive events specifically contingent on either the cyr1 or the gpb2 mutations, reflecting the phenotypic similarity of the cyr1 and gpb2 genotypes[4,9,20]. Nevertheless, that does not preclude that with a larger sample size we may have observed genotype-specific adaptive responses for cyr1 and gpb2. Finally, future studies measuring the fitness effects of genotypes with adaptive mutations in combinations that were not observed would be informative on the potency of historical contingency in our system.

We note that isolation of clones, here and in our earlier study[4], undoubtedly enriched for the highest fitness lineages. As a result, several open questions remain: Are the mutations that emerged as adaptive during the 2nd step not at all adaptive during the 1st step? Are they adaptive but not adaptive enough to be seen by selection at the bottleneck we are applying? Or are they adaptive enough to be seen by selection but not sufficiently so to be detected given our sampling method? Thus, a potential effect of clonal interference, whereby the most fit mutations are selected first, is a possible source of the observed diminishing returns. More work is needed to tease apart the effects of clonal interference and antagonistic interactions between the 1st and 2nd adaptation step mutations by measuring the effects of specific adaptive mutations on backgrounds of different fitness.

A main limitation of our system stems from the fact that the evolutionary condition includes constant mixing of a large population. Consequently, clonal interference has a pervasive influence on the outcome since multiple beneficial mutations enter the population each generation and each adapted lineage is in direct competition with all other adapted lineages. Even when we restrict analysis to short timescales, prior to clonal interference taking a toll on the population's genetic diversity[41], as in the present study, we are only able to detect and characterize the most prevalent lineages, those that arose the earliest, and/or harbor the highest fitness mutations. While lineage tracking via molecular barcoding has improved our detection threshold to identifying lineages with frequencies as low as 1 in a million (Supplementary Fig. 2) we are still limited to lineages with fitness coefficients bigger than 0.02 per generation. Nonetheless, very low fitness lineages do not contribute substantially to the evolutionary dynamics in this kind of experiment[4]. That is further supported by experiments in E. coli showing that weakly adaptive mutations can behave as if they are neutral in well-mixed environments, resulting in evolutionary stalling when higher fitness events are not available[70]. However, it is expected that evolutionary outcomes in less competitive environments (for example in structured environments where lineages mainly compete with their neighbors[71,72]) strongly depend on neutral to low fitness variation as well. Additionally, clonal interference could skew the DFE inferred from the evolving population. As population mean fitness increases, the effect of clonal interference becomes more significant; newly emerged adaptive mutations are effectively less beneficial and less likely to establish in the population, making it difficult to accurately estimate lineages' fitness at this point. By isolating thousands of lineages from the evolving population and competing them with the wild-type ancestor, we are able to assess to what extent clonal interference affects the fitness estimates. We find that the fitness remeasurements are largely consistent with the fitness estimates from the evolution data (see Supplementary Fig. 8), which is also consistent with our prior experience (see Fig. 2D–F in ref. [4]). Thus, we conclude that our DFE is not (or at least not obviously) skewed by clonal interference.

Nevertheless, within the limits of detection, our results provide clear evidence for the role of evolutionary history in shaping selection during subsequent adaptation steps. This suggests that after acquisition of even a single adaptive mutation the selective pressure a population experiences can change, even in the absence of environmental perturbation (though we acknowledge that a different clone growing in the same media may in fact perturb the environment). Here, the first adaptive change might be considered to be in direct response to the environmental condition, where adapted lineages modified their nutrient signaling pathways to respond to an environment that predictably undergoes glucose feast and famine[4,20]. Adaptive changes in pleiotropic genes (such as those that regulate nutrient signaling) may include non-adaptive or even maladaptive side effects. Thus, the set of second adaptive mutations may be constrained to adjust for pleiotropic consequences of the first, compensating for suboptimal changes to the cellular network. This suggests that fine-tuning of the same pathway may be minimally beneficial in a majority of cases, compared to responses that adjust different pathways. Specific to our experiments, extensive work (cited above) suggests that growth optimization comes with a cost in stress responses, and as a result, 2nd-step adaptation strategies targeting modification of stress responses may be contingent on the nature of the adaptive strategy caused by the 1st step. This shift in adaptive strategy may underlie the observation that the 2nd adaptive step was more often due to presumptive GOF mutations.

## Methods

**Strains and strain handling**. All strains used are S288C derivatives, which were evolved and characterized previously[4,20,44] (Table 1). Yeast strains and pools were saved as glycerol stocks at −80 °C. Yeast transformations were performed by a lithium acetate/PEG method[73]. Strains are available upon request.

**Yeast growth media and growth cycle**. Evolutionary and fitness remeasurement conditions matched those used earlier[4,44]. Briefly, M3 medium, consisting of 5× Delft medium[74] with 4% ammonium sulfate and 1.5% dextrose, was used. Serial batch cultures were conducted by growing cells in 100 mL M3 medium in 500 mL Delong flasks (Bellco) at 30 °C and 223 RPM. Yeast was grown for 48 h between transfers and for each new cycle 400 µL of the grown culture (~8 × 10⁷ cells) were used as inoculum for the new culture, resulting in a 1:250 bottleneck.

**Construction and characterization of the founder strains of the evolutions**. The prelanding pad strain (SHA118, Table 1) that is receptive to barcoding[44] was transformed with a galactose-inducible HO-containing plasmid. The strain diploidized upon exposure to galactose and a diploid clone was sporulated and dissected; a *MATa* derivative was isolated (GSY5375, Table 1) and saved for subsequent crosses with the evolved clones. Loss of the HO-containing plasmid was verified by absence of growth on appropriate selective medium. Strains GSY5481, GSY5128, and GSY5153, derived from evolution under glucose limitation and previously characterized[4,20,44] (Table 1), were backcrossed twice to GSY5375. Competitive fitness of segregants and evolved parents was assayed in triplicate compared to a fluorescent derivative of the ancestor, as in ref. [4], with the following modifications: to increase throughput the assays were performed in 5 mL cultures in tubes incubated in a roller drum, instead of 100 mL cultures in flasks. As a result, the fitness estimates deviate from those previously reported[4] (Supplementary Fig. 1, Supplementary Table 1). Additionally, since the derived strains do not contain a barcode (which reconstitutes a *URA3* gene), they require uracil, so the fitness assays were performed in M3 medium supplemented with uracil. All segregants were genotyped for the variant of interest by amplification of the respective locus and Sanger sequencing. The oligos used for genotyping are shown in Supplementary Table 3.

**Barcoding**. Strains GSY6701, GSY6702, and GSY6703 (Table 1) were transformed with a low and high complexity barcode, consecutively. These strains have the *YBR209w* locus replaced with the prelanding pad (corresponding to strain SHA118[44]). The low complexity barcode was derived by PCR amplification of part of the L001 plasmid library, containing the lox66 site, the DNA barcode, the artificial intron, the 3′ half of *URA3*, and *HygMX*[75]. The fragment was amplified with primers BC_F-DY and BC_R1-DY (Supplementary Table 3), from 12 ng of L001 library in a 50 µL reaction with PrimeSTAR (TAKARA, Mountain View, CA)

using the following conditions: hot start, initial denaturation at 98 °C for 2′, 30 cycles of 98 °C for 30″, 55 °C for 15″ and 72 °C for 3′, and final extension at 72 °C for 10′. The product was purified with the QIAquick PCR purification kit (QIA-GEN, Germantown MD), transformed into each of GSY6701, GSY6702, and GSY6703 and successful transformants were selected on YPD + Hygromycin. Single transformants were further transformed with the high complexity library (pBAR3)[44] with the following modification: after transformation the cells were grown on liquid YP + 2% galactose for ~16 h for Cre recombinase induction prior to selection on SC-ura plates with 2% glucose. Cell growth was estimated by cell counting immediately after transformation and before plating. The number of unique transformants was estimated by plating a dilution on selective medium and correcting for growth. After 1 day of growth on selective medium the transformants were pooled and saved as glycerol stocks at −80 °C (high complexity subpools with a common low complexity barcode). The final founding pools for the evolutions were constructed by pooling high complexity subpools to an estimated total of ~700,000 unique transformants per initial clone.

**Evolution experiments**. Evolution experiments were conducted under identical conditions to those that gave rise to our adapted ancestors[44]. Briefly, 10⁸ cells of each of the founding populations were used to inoculate 100 mL of SC-ura, 2% dextrose, supplemented with hygromycin in 500 mL Delong flasks (Bellco). The cells were grown for 24 h at 30 °C and 223 RPM, the end of which was considered generation 0 of the evolution experiment. 400 µL of the initial culture were used to inoculate M3 medium in duplicate, as described in the 'Yeast Growth Media and Growth Cycle' section. The evolution experiments were conducted for a total of 20 transfers, corresponding to approximately 160 generations. Prior to each transfer the medium was prewarmed at 30 °C for 1 h. For each timepoint, 2 × 1 mL aliquots were saved as glycerol stocks at −80 °C and the rest of the culture was spun down, resuspended in 5 mL sorbitol buffer (0.9 M sorbitol, 100 mM Tris pH 7.5, 100 mM EDTA), aliquoted in Eppendorf tubes (~1 mL each), and saved at −20 °C to be used for genomic DNA and barcode library preparations.

**Clone isolation**. Individual clones were sorted at the Stanford Shared FACS facility either from all timepoints (one or two 96-well plates each) for ploidy determination, or from selected timepoints (10 plates each of the following timepoints: *cyr1* evolution, replicate 1, timepoint 20 (generation 160), *gpb2* evolution, replicates 1 and 2, timepoint 13 (generation 104) and *tor1* evolution, replicate 1, timepoint 12 (generation 96)) for fitness remeasurements, ploidy determination and whole-genome sequencing.

**Ploidy assay**. Ploidy was determined with a high-throughput benomyl-based assay[4]. Clones archived in 96-well format were grown in deep-well plates to saturation at 30 °C without shaking. The saturated cultures were mixed with pipetting and subsequently pinned onto YPD agar rectangular plates containing 20 mg/ml benomyl (prepared as a DMSO solution) or DMSO (control). The plates were grown at 25 °C for 2 days and then imaged. Clones with inhibited growth on the benomyl medium were identified as diploids. Clones with inhibited growth on the control medium were excluded from analysis. Clones that grew on both media were identified as haploids.

**Genomic DNA and library preparation for barcode lineage tracking**. Genomic DNA was prepared as follows. An aliquot of cells stored at −20 °C was thawed at room temperature. The cells were spun down, washed once in water, resuspended in 400 µL lysis buffer (0.9 M sorbitol, 50 mM Na phosphate pH 7.5, 240 µg/mL zymolase, 14 mM β-mercaptoethanol) and incubated at 37 °C for 30 min. After the incubation, 40 µL 0.5 M EDTA, 40 µL 10% SDS and 56 µL 20 mg/mL proteinase K (Life Technologies 25530-015) were added consecutively, with brief vortexing after each addition, and the samples were incubated at 65 °C for 30 min. Samples were then cooled on ice for 5′, 200 µL of 5 M potassium acetate were added, and the samples were mixed by shaking, then incubated on ice for an additional 30 min. Following incubation, the samples were spun down at full speed in a microcentrifuge for 10 min, and the supernatant was transferred to a new tube with 750 µL isopropanol and was let to rest on ice for 5 min. The precipitated nucleic acid was spun down full speed in a microcentrifuge for 10 min and washed twice with 70% ethanol. After the second wash the nucleic acid was let to dry completely and then it was resuspended in 50 µL 10 mM Tris pH 7.5. Overnight incubation at room temperature or short incubation at 65 °C sometimes was necessary for complete resuspension. RNA was digested with the addition of 0.5 µL 20 mg/mL RNase A (Fisher Scientific, Waltham MA) and incubation at 65 °C for 30 min.

A two-step PCR protocol was used to amplify the barcoded locus (see Supplementary Table 3 for primers, same as used previously[4]). The first amplification was conducted using OneTaq 2X Master Mix (NEB, Ipswich MA), a total of 6 µg genomic DNA and a limited amount of primers in 6 × 50 µL reactions with the following composition: 1X OneTaq Mix, 50 nM each forward and reverse primer, 2 mM MgCl₂, 20 ng/µL gDNA, in the following conditions: hot start, initial denaturation at 94 °C for 10′, 3 cycles of 94 °C for 3′, 55 °C for 1′ and 68 °C for 1′, and final extension at 68 °C for 1′. The 6 reactions were combined, purified using the QIAquick PCR purification kit (QIAGEN, Germantown MD) and eluted into

30 μL EB buffer. All the eluate was used as template in a single 50 μL 2nd reaction, with the following composition: 0.5 μL Herculase II fusion DNA polymerase (Agilent, Santa Clara CA) per 50 μL reaction, 1×Herculase buffer, 1 mM dNTPs, and 500 nM each of PE1 and PE2[4], and was amplified in the following conditions: hot start, initial denaturation at 98 °C for 2′, 20 cycles of 98 °C for 10″, 69 °C for 20″ and 72 °C for 30″, and final extension at 72 °C for 1′. Barcode libraries were pooled isostoichiometrically and sequenced on an Illumina NextSeq 550.

**DFE/mutational fitness spectrum μ(s) inference.** Lineage tracking from barcode sequencing was performed as described[44], using code at https://github.com/Sherlock-Lab/Barcode_seq/blob/master/bartender_BC1_BC2.py with minor modifications. Briefly, after extraction of the unique molecular identifiers (UMIs, randomers used for identifying duplicates introduced during PCR amplification), and both low and high complexity barcodes from the sequencing read, low complexity barcodes were clustered against their expected sequences, whereas the high complexity barcodes were pooled across all libraries and clustered with bartender (v1.1)[76]. The updated reads and the UMIs were used to derive raw barcode counts, which were assembled into the raw count lineage trajectories. Low coverage timepoints and barcodes that appeared in only a single timepoint (considering replicate evolutions) or had no reads at timepoint 0 were excluded from subsequent analysis. The included timepoints and the number of reads and barcodes per timepoint are shown in Supplementary Data 1. Filtered raw count lineage trajectories are provided for each replicate evolution (Source data for Supplementary Fig. 2, "Lineage trajectory counts").

Using the lineage frequency changes over time, lineages' fitness per generation (s) and establishment time (tau) were estimated as in ref. [44]. Lineages with reads between 20–30 at each timepoint were treated as neutral and were used to estimate population mean fitness. Lineage tracking data from generation 0 to generation 136 were used for fitness inference in all evolutions, except for *gpb2* evolution replicate 1, for which we only had adequate data up to generation 120. Lineage tracking data for the ancestor up to generation 112 and 96, for replicates 1 and 2, respectively, were used for fitness inference as in ref. [44]. The generations chosen are the times at which adapted lineages have reached a sufficient frequency in the population, while the majority of such lineages theoretically carry a single beneficial mutation.

Mutations can occur during the barcoding process and prior to the onset of the experiment, some of which can be beneficial in the evolutionary condition[44]. To characterize the mutational rate during the evolution, lineages with such pre-existing mutations were removed from fitness inference. The following two criteria were used to define lineages with pre-existing mutations: (1) being adaptive in both evolutionary replicates and (2) having an establishment time < −2/s in at least one replicate.

Mutation rates (μ(s)*ds, defined as the mutation rate per generation per cell for mutations with fitness within a range [s, s + ds]) in different fitness intervals were calculated using equation 101 in ref. [44]:

$$\mu(s)ds\left[1 + s\ln\left(N_f\mu(s)ds\right)\right] = f(d(s), t)\frac{s}{e^{s\bullet t}} \quad (1)$$

where $ds = 0.002$ is the fitness interval considered, $\mu(s)$ the mutation rate within a specific fitness interval $[s, s + ds]$, defined as the fitness-dependent probability density function of the mutation rate, $N_f = 10^{12}$, the approximated largest size the population has reached during the barcoding process, and $f(d(s), t)$ the summed frequency of lineages whose fitness are within the interval $[s, s + ds]$ at generation $t$. The error of the estimated $\mu(s)$ is:

$$\sqrt{\frac{\mu(s)\bullet ds}{N}} \quad (2)$$

where $N = 6\bullet10^8$ is the approximated effective population size during evolution.

Note that the barcode sequencing coverage of the ancestor evolutions was ~10–20× higher than those of *cyr1*, *gpb2*, and *tor1* evolutions (Supplementary Data 1, also Table 2 in Supplementary information in ref. [44]). To avoid biases introduced by sequencing coverage differences, we down-sampled the ancestor sequencing data to a depth comparable to those of the *cyr1*, *gpb2*, and *tor1* evolutions: $2 \times 10^7$ at time 0 and $3 \times 10^6$ at the rest of the timepoints. Fitness was inferred before and after down-sampling. Lineages with 5–10 reads were treated as neutrals to infer the population mean fitness (vs 20–30 used in the full datasets).

The theoretical expectations on our detection threshold were set as in ref. [44]. The fitness detection threshold was determined by our definition of neutrality. We treated low abundance lineages as neutrals and used the decay of those lineages over time to infer population mean fitness. The resolution of our study cannot detect adaptive lineages with a fitness < 2% due to clonal inference and the lineage size limit (see Supplementary Fig. 36 in ref. [44]). Certain mutations with fitness < 2% can establish in the population without being detected. However, such mutations never reach large sizes and will not dominate or have much of an impact on population dynamics. In particular, during the 1st step evolutions, population mean fitness reached 0.02 at generation 90. Due to clonal interference, mutations with s = 0.02 have to arise before generation 40 (90–1/s) in order to establish. To be detected, such mutations have to arise 60 generations before the start of the evolution experiment (from equation 77 in ref. [44]; $90 - 1/s\bullet\ln(n_e\bullet s)$, where $n_e = 1000$ cells/lineage represents the lineage size). Thus, mutations of such effect that

occur immediately at generation 0, are nearly impossible to be detected unless this group of mutations has a very large mutational target size or a very high mutation rate. This is illustrated in detail in the Supplementary material in ref. [44], section 11.1, page 54.

**Barcode determination of isolated clones.** To identify the barcodes of the isolated clones in 96-well plates, we employed a 2- (column, row) or 3- (column, row, plate) dimensional pooling strategy, inspired by ref. [77]. Briefly, we arranged 20 plates per batch into a 4 columns × 5 rows plate matrix and constructed 48 column pools from clones out of 40 wells each and 40 row pools from clones out of 48 wells each. For the second batch we included semi-redundant half-plate pools (40 pools from clones out of 48 wells each) to increase the successful barcode recovery rate. We pooled our samples after cell growth and prepared barcode libraries for Illumina sequencing. Barcodes were recovered for each well at a rate of ~90%, which was somewhat dependent on the barcode diversity of the sampled timepoint (identical barcodes in multiple wells makes it more challenging algorithmically to match barcodes to wells).

**High-throughput fitness measurements and analysis**
*Pooling of clones.* Clones isolated from the evolutions were pooled together for high-throughput fitness assays. We used a multi-pronged pinner to take clones from frozen stock and pin them into a set of 96 deep-well plates with 700 μL YPD medium in each well. Cells were grown at 30 °C for 2 days to reach saturation without shaking. 500 μL of 50% glycerol were added into each well using a multichannel pipette. 1 mL of the mixture from each well was pooled, and the final pool was mixed and aliquoted into 2 mL Eppendorf tubes, which were stored at −80 °C for future fitness measurements.

*Preculture.* Each replicate fitness experiment was initiated with a 1 mL frozen aliquot of the pooled cell culture, thawed at room temperature, and inoculated into 15 mL M3 in a 500 mL Delong flask. The culture was grown at 30 °C and 223 RPM overnight for cell propagation. 400 μL of the overnight culture were inoculated into 100 mL of fresh M3 medium and precultured at the standard condition for 2 days.

A derivative of the ancestor carrying a restriction site in the barcode region was used to compete with the pool of evolved clones for fitness measurements[4]. The ancestor clone was resurrected from frozen stock onto M3 agar plates and grown for 2 days until colonies were visible. A single colony was inoculated into 3 mL of M3 medium and grown for 48 h (30 °C in a roller drum). 400 μL of that culture were used to inoculate precultures (100 mL M3 medium in 500 mL Delong flasks, 223 RPM 30 °C).

*Competition.* Fitness assays were conducted by mixing the pooled preculture with the ancestor preculture in a 1:9 ratio (time 0) and growing the resulting population for four successive growth cycles (timepoints 1, 2, 3, and 4), under the evolutionary condition. At the end of each cycle, 400 μL cell culture were inoculated into 100 mL fresh media to start the next cycle. Cells were collected at time 0, and at the end of each of the four growth cycles. The cell pellet from each sample was resuspended in 5 mL sorbitol solution (0.9 M sorbitol, 0.1 M Tris-HCL pH 7.5, 0.1 M EDTA pH 8.0), aliquoted into 2 mL Eppendorf tubes and stored at −20 °C. Three technical replicates were performed per fitness assay. Genome extraction, barcode amplification, and Illumina sequencing were conducted for each sample (timepoint and replicate).

*Genomic DNA sample preparation.* Genomic DNA was isolated and treated as described in the "Genomic DNA and library preparation for barcode lineage tracking" section.

*Fitness estimation.* DNA barcodes were sequenced on an Illumina NextSeq 500/550 platform and their abundances were used to estimate lineages' frequencies in the population, as previously described[4]. Fitness estimates were conducted for all clones against the neutrals from the wild type evolution and for clones derived from each ancestral genotype separately against the neutrals of the specific genotype. The source code for computing these fitness estimates can be found at https://github.com/barcoding-bfa/fitness-assay-python. We ran two iterations of the script. First, we used all barcode counts as input and recovered fitness estimates and barcodes that were likely to be neutral. Barcodes identified by the first iteration were associated with their physical position on the 96-well plates in frozen stock, and the ploidy of the clones they represent. For the second iteration, apart from the barcode counts, a list of specifically haploid neutral clones was also provided (this is an optional argument of the fitness estimation algorithm). Fitness estimates from the 2nd run were used for further analysis. Final fitness estimates were calculated by inverse variance weighting of estimates from all three replicates.

**Calculation of diploidization rates.** Diploidization rates were estimated for each genotype and replicate evolution using information from both the fitness remeasurements and lineage tracking data. First, we identified pure diploids based on fitness remeasurements and ploidy assays (see "Fitness gains of isolated adaptive

clones from the 2nd-step evolutions tend to be smaller than in clones from 1st-step evolutions" section in Results for details). Second, we identified which of these diploids were "pre-existing diploidy lineages" (see "DFE/Mutational fitness spectrum μ(s) inference" section for details), which are beneficial in both evolution replicates, suggesting that their diploidization events likely happened before evolution. Thus, the "pre-existing diploidy lineages" is a group of diploids whose ploidy has been experimentally verified but does not contribute to the estimate of diploidization rate during evolution. In addition, "pre-existing diploidy lineages" whose fitness was estimated to be larger than or equal to 0.1 per generation from the lineage tracking data, were also excluded, in order to avoid diploids that had acquired additional mutation(s).

Diploids arising during evolution were identified via their fitness values because not all evolving lineages were assayed for their ploidy. Lineages that do not carry pre-existing mutations and have a similar fitness to those of the "pre-existing diploidy lineages" were characterized as diploids that emerged during evolution. Specifically, using the lineage tracking data, we estimated the mean fitness of "pre-existing diploidy lineages" and its corresponding 95% CI for each evolution replicate. Lineages whose fitness fell into this 95% CI were classified as diploids and were used to estimate the diploidization rate for each replicate evolution. Note that the mean fitness of diploids and its 95% fitness CI were estimated using a small group of curated "pre-existing diploidy lineages". Thus, lineages whose fitness fall into the CI are likely only a subgroup of diploids that arose during evolution and our calculation results in a conservative and therefore likely underestimate for the diploidization rate per genotype and replicate evolution.

**Genome-wide sequencing library preparation**. Clones selected for sequencing were grown in 500 μL YPD in 96 deep-well plates for two days at 30 °C without shaking. 400 μL of saturated cell culture were used for DNA extraction with the Invitrogen PureLink Pro 96 Genomic DNA Kit (Catalog no. K1821-04A) in a 96-well format. Libraries were prepared and multiplexed with Nextera technology, and a high throughput protocol[78]. Samples were sequenced on an Illumina HiSeq 4000 with 2 × 150 bp paired end reads.

**Variant calling**. SNP, small indel, and structural variants were called for 105 clones using Sentieon Genomic Tools Version 201711.02[79], as follows. FASTQ files were trimmed using cutadapt version 1.16[80] and trimmed reads were mapped to the S. cerevisiae S288C reference genome R64-1-1 (https://downloads.yeastgenome.org/sequence/S288C_reference/genome_releases/) using bwa[81]. Mapped and sorted reads were then used for the variant calling. Variants were further annotated using snpEff and SNPSift[82]. The source code for variant calling and annotation can be found at https://github.com/liyuping927/DNAscope-variants-calling.

**Variant filtering**. To eliminate false positive variants, we applied the following filters. First, variants from lineages with an average genome-wide coverage <10, and all mitochondrial variants were filtered out. Second, variants in *FLO1* and *FLO9* genes were filtered out due to poor alignment in both genomic regions. Third, variants present in more than five clones and at least two genetic backgrounds out of *cyr1*, *gpb2* and *tor1* mutants, they were likely present in the common ancestor and were filtered out. Fourth, variants with a quality score <150 and only occurring in one clone were filtered out. Locus alignment against the reference genome was visually inspected to assess variants present in more than one clone, but with a quality score <150 in at least one of them. Provided that the implicated loci were well-covered and not in highly repetitive regions, the variants were considered *bona fide* regardless of their quality score. Otherwise, they were discarded in all clones where they occurred. Lastly, we manually verified variants that passed the above filtering by inspecting the corresponding loci alignments against the reference genome and further filtering out false positives, typically occurring in highly repetitive or poorly sequenced regions.

**Reporting summary**. Further information on research design is available in the Nature Research Reporting Summary linked to this article.

## Data availability
Sequencing data that support the finding of the study are deposited in Short Read Archive under Bioproject ID PRJNA641174. The rest of the data are available in the main text and Supplementary Tables. All strains are readily available from authors upon request. Source data are provided with this paper.

## Code availability
The code for fitness calculations from lineage tracking is described in ref. [44]; the code for fitness calculation from remeasurement experiments is available at https://github.com/barcoding-bfa/fitness-assay-python. The code for the barcode sequencing data processing can be found at https://github.com/Sherlock-Lab/Barcode_seq/blob/master/bartender_BC1_BC2.py. The code for variant calling from whole-genome sequencing data can be found at https://github.com/liyuping927/DNAscope-variants-calling.

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

## Acknowledgements
The authors wish to thank Lucas Hérissant for help with FACS sorting and the benomyl assay, Jamie Blundell for help with data analysis, and Frank Rosenzweig and Sasha Levy for comments on the manuscript. The study was funded by NASA grant NNA15BB04A and NIH grant R35 GM131824 to GS.

## Author contributions
G.S. conceived the study. D.A. and Y.L. designed and performed the experiments and analyzed the data. D.A., Y.L., and G.S. wrote the manuscript.

## Competing interests
The authors declare no competing interests.
