## [Peer Review File · Nature Communications]

Changes in the distribution of fitness effects and adaptive mutational spectra following a single first step towards adaptationReviewers' Comments:

Reviewer #1:

Remarks to the Author:

Review of Aggeli et al. for Nature Communications

What sequence of genetic events gives rise to adaptation, and when are these events predictable, in terms of their molecular basis and magnitude? Aggeli et al. applied a recently-developed method to address these persistent questions. Previously, adaptive variants were identified in yeast populations evolving under glucose limitation, using a lineage tracking technique. In this study, the authors essentially repeated the previous experiment using genetic backgrounds that had already obtained a given "first step" adaptive mutation.

Beneficial mutations that arose in the second step were somewhat predictable, often appearing among the three genetic pathways originally identified, but not in the same pathway as that of the first step mutation. In addition, ploidy changes were very common, as in the preceding experiment. However, the authors also report evidence for a shift in the spectrum of adaptive variants from the first step to the second step, in terms of both the molecular characteristics and fitness spectrum of beneficial mutations.

The use of DNA barcoding is a novel approach to understanding historical contingency in evolution, but this study relies more on observation than on theoretical predictions. This approach confirms the common observation that adaptation takes place with diminishing returns, but it isn't very clear what insight the barcoding approach has added in this respect. The authors suggest that diminishing returns could be due to both smaller fitness effects of second-step mutations and smaller numbers of available mutations in the second step, i.e., a smaller mutational target size (line 616). However, the ability to detect weakly beneficial mutations using this approach is limited, with no data available for $s < 2\%$. It is therefore not clear that these data can distinguish between a shift in the height of the distribution of fitness effects and a shift in the shape of the distribution; alleles that may have had detectable effects in the first step may have weaker, non-detectable effects in the second step, confounding the beneficial mutation rate with mutation effect sizes. The theoretical expectations on this topic should be outlined, along with limitations of the data.

A specific class of beneficial mutation--diploidization--creates a complication for these analyses, because it arises often enough to dominate the distribution of fitness effects, evidently creating a distinct peak (line 589). Despite the fact that diploidization of haploid yeast was observed in the preceding study of these populations (and is commonly observed in yeast evolution experiments using haploids) there is no mention of the issue in the introduction. This is surprising given that the appearance of diploids would be an obvious prediction to make. The possible fitness benefits of diploidy are interesting in their own right, and it would be great to see estimates of diploidization rates and effects along with confidence intervals.

The authors indicate that one of the benefits of their study is the short time scale in terms of generations, but it's not obvious why evolutionary dynamics on this timescale are more relevant than longer timescales. If this were a natural population we would expect continued clonal competition among adaptive genotypes, as well as recombination among genotypes, which could presumably result in a single best genotype on a longer timescale. In other words, evidence of historical contingency at short time scales may not be sufficient to demonstrate that historical contingency is important at moderately longer timescales. Indeed, even at the short timescale of the first few adaptive changes studied here, the data seem to offer a fair amount of support for convergent adaptive pathways rather than contingency, with mutations in two particular pathways, plus diploidization, being the most common. There is therefore some risk that these findings would add to, rather than clarify, the mixed evidence on historical contingency described in the Introduction. It would be good to see a nuanced discussion of these issues and caveats as a guide for future research.

Additional comments:

Diploidization of the genome presumably doubles the barcode sequence, creating a difference between the frequency of a barcode and the frequency individuals with the corresponding genotype. Is there a need to account for this in general, and specifically when estimating the fitness effects of diploidy?

The negative correlation between ancestral fitness and highest evolved fitness (lines 491, 586, Fig. S6) is consistent with diminishing returns, but not likely to be statistically significant with just four data points. A rank correlation would be more appropriate, but still won't be formally significant given the sample size. Perhaps there are more powerful ways to test for this association, e.g., by making use of randomization to generate an appropriate null distribution.

Line 600. Here and elsewhere I think it's important to clarify that this does not refer to mutation rates in general, but rather the rate that mutations arose that were beneficial enough to be detected.

The abbreviation UMI seems to appear only twice (lines 231 and 234), and could be avoided, but should otherwise be defined.

Reviewer #2:

Remarks to the Author:

>> Summary

The authors conducted a comprehensive analysis of the 2nd mutations arising from three genetic backgrounds differed by one adaptive mutation (i.e. 1st mutation). They found a significant difference between the overall mutation spectrums of the adapted and wild-type (WT) ancestors. Among the three adapted ancestors, they found mutations arising from the two Ras/PKA pre-adapted ancestors more similar than those from the TOR/Sch9 pre-adapted ancestor. The authors discussed the implications of their findings in historical contingency and the frequently seen diminishing trend of adaptation.

>> Major Comments

Changes in the spectrum (i.e. fitness effect, probability, molecular type ...) of 1st and 2nd mutations in adaptation is the major observation of this work, which is both interesting and important in the study of DFE. This finding could result from (1) clonal interference, enriching the strongest beneficial mutation in the first step or (2) antagonistic interaction between the 1st and 2nd mutations. Unfortunately the resolution of fitness assays and the analysis presented by the authors could not make a clear distinction or estimation of the relative contribution between of the two causes. Such limitation compromises the value and novelty of this work.

On the other hand, while this study shows that the 2nd mutations arising from the TOR/Sch9 and Ras/PKA backgrounds are somewhat different, the experimental design and results do not seem suitable to address historical contingency. What the authors observed in such short time frame might reflect just a transient alteration of the temporal order of mutational events and may have limited impacts on the long-term evolutionary trajectory. Particularly, it seems unnecessary to make a big deal about seeing TOR/Sch9 mutations more likely to arise in the Ras/PKA pre-adapted ancestor, and vice versa. The authors' prior work already shows the predominance of these two mutation types in the first-step mutation spectrum and suggests that their beneficial effects do not appear to cancel out.

If the authors could run the evolution experiment longer, say 1000 generations, the three types of adapted ancestors might end up acquiring a similar set of mutations. Such result, if occurs, would indicate the three lineages following similar evolution trajectories, and as such, no historical

contingency. The authors could address this concern by (1) evolving their populations longer or (2) referring to the results of prior evolution experiments involving similar strains and growth conditions.

Line 560-562: Results presented here contradict the message in Line 98 and Line 115. The title of this section should be rephrased as well.

>> Minor Comments

Line 37-41: The interpretation of results is a circular argument.

Line 53: Two very first studies of diminishing returns epistasis are missing here - Chou (2011) and Khan (2011).

Line 388: RAS or Ras. The style is not consistent throughout.

Figure 2: Does "density" mean the probability density function of the mutation rate? This term is not defined. Moreover, it would be nice to show the fitness distribution of the two major types of beneficial mutations (i.e. TOR/Sch2 & Ras/PKA)

Line 478-479: I am not able to see this in Fig. 3.

Line 491: Report P value here.

Line 535: Table or table? Style consistency.

Line 611-613: Recent studies of global fitness landscapes, such as Diss (2018), Poelwijk (2019), Bendixsen (2019), and Kuo (2020), are worth comparing here.

Line 617: Correct font style.

Line 631-632: Discussion of sign epistasis appears out of blue.

Line 665-666: Add "in the subsequent step."

Reviewer #3:

Remarks to the Author:

In this paper Aggeli et al., study the distribution of fitness effects during the second step of adaptive evolution in an experimental evolution regime. Starting with three strains that were recovered from a prior evolution experiment containing adaptive mutations in TOR1, CYR1 and GPB2 the authors introduced a complex library of random molecular barcodes into each strain. They then performed experimental evolution using serial batch dilution. Using the dynamics of the barcoded lineages they estimate the distribution of fitness effects for mutations acquired during the second step of adaptive evolution. Using whole genome sequencing of lineages they identify the molecular basis of the second adaptive mutation. They conclude that their results are consistent with diminishing return epistasis.

This is an interesting and well performed study. Although the conclusion is consistent with prior studies, it provides additional evidence for the generalizability of global epistasis in adaptive evolution, which is of importance to the field. Moreover, there are several interesting observations regarding the functional relationship between targets of adaptive mutations following the acquisition of a beneficial mutation that are of interest to the field. Prior to publication, the authors should consider the following points:

-In the introduction the authors state that they have characterized the rate of adaptation, but I do not

find this in the paper. I would expect to see a quantification of the rate of increase in population fitness. Is this possible to infer with the data? This would be an interesting result to understand how initial population fitness impacts the rate of adaptation

-There is no data showing the benomyl assay for detecting diploids. I think that some representative result should be presented - even if it has been shown in a prior publication - and an explanation as to why a more sensitive assay such as DNA staining and FACS wasn't used.

-The initial beneficial mutations are referred to as "presumptive gain of function", but there is no evidence that this is the case and this terminology implies that the activity of these pathways is increased, which has not been shown. Why not just call them beneficial mutations? Or demonstrate that they are indeed gain of function using reporter, or phosphorylation, assays. Moreover, two of the studied mutations are not loss of function mutations which conflicts with the authors' statements about the prevalence of loss of function mutations in the first adaptive step.

-Figure 2 reproduces the same wildtype data three times. Does plotting the three different distributions of second step mutations on the same plot make it possible to compare between the different backgrounds? Perhaps it would be too cluttered, but it would be useful to see the extent to which the DFE differs between the three backgrounds. The error bars may be unnecessary for this visualization.

-Figure S7 needs a quantification of the correlation

Response to Reviewer Comments

We thank the reviewers for their comments – below we provide a point-by-point
response to each of the comments. Our responses are highlighted, while the reviewers'
comments are italicized.

*Reviewer #1 (Remarks to the Author):*

*Review of Aggeli et al. for Nature Communications*

*What sequence of genetic events gives rise to adaptation, and when are these events*
*predictable, in terms of their molecular basis and magnitude? Aggeli et al. applied a*
*recently-developed method to address these persistent questions. Previously, adaptive*
*variants were identified in yeast populations evolving under glucose limitation, using a*
*lineage tracking technique. In this study, the authors essentially repeated the previous*
*experiment using genetic backgrounds that had already obtained a given “first step”*
*adaptive mutation.*

*Beneficial mutations that arose in the second step were somewhat predicable, often*
*appearing among the three genetic pathways originally identified, but not in the same*
*pathway as that of the first step mutation. In addition, ploidy changes were very*
*common, as in the preceding experiment. However, the authors also report evidence for*
*a shift in the spectrum of adaptive variants from the first step to the second step, in*
*terms of both the molecular characteristics and fitness spectrum of beneficial mutations.*

*The use of DNA barcoding is a novel approach to understanding historical contingency*
*in evolution, but this study relies more on observation than on theoretical predictions.*
*This approach confirms the common observation that adaptation takes place with*
*diminishing returns, but it isn't very clear what insight the barcoding approach has*
*added in this respect. The authors suggest that diminishing returns could be due to both*
*smaller fitness effects of second-step mutations and smaller numbers of available*
*mutations in the second step, i.e., a smaller mutational target size (line 616).*

We agree that our study is empirical rather than based on theoretical predictions.
However, we argue that use of barcodes provides the following advantages: Barcoding
1) is necessary to generate the distribution of fitness effects (DFE), 2) allowed us to
determine a point in each evolution at which there would be sufficient adaptive lineages,
3) allowed us to subsequently perform pooled fitness assays on independent isolated
lineages, and 4) allowed us to whole genome sequence lineages that we know are
independent. Together, these benefits of barcoding allowed us to deeply characterize
the beneficial mutational spectra and the fitness effects of those mutations, providing
substantial insight into the potential causes underlying diminishing returns/historical
contingency. We have modified the introduction to better emphasize the advantages of
barcoding in understanding the phenomena of historical contingency and diminishing
returns, in lines 83-90.

*However, the ability to detect weakly beneficial mutations using this approach is limited,*
*with no data available for $s < 2\%$. It is therefore not clear that these data can distinguish*
*between a shift in the height of the distribution of fitness effects and a shift in the shape*
*of the distribution; alleles that may have had detectable effects in the first step may*
*have weaker, non-detectable effects in the second step, confounding the beneficial*
*mutation rate with mutation effect sizes. The theoretical expectations on this topic*
*should be outlined, along with limitations of the data.*

**The source of the detection threshold:** Fitness estimates use low abundance
lineages as neutrals and their decay over time to infer population mean fitness. The
resolution of our study cannot detect adaptive lineages with a fitness $< 2\%$ due to clonal
interference and the lineage size limit (see Supplementary Figure 36 in Levy *et al.*,
Nature 2015). Certain mutations with fitness $< 2\%$ can establish in the population without
being detected. However, such mutations never reach large sizes and will not dominate
or have much of an impact on population dynamics.

**Theoretical expectations** (Levy *et al.*, Nature 2015): During the 1st step evolutions,
population mean fitness reaches 0.02 at generation 90. Due to clonal interference,
mutations with $s=0.02$ have to arise before generation 40 (equation: $90-1/s$) in order to
establish in the population. In order to be detected, such mutations have to arise 60
generations *before* the start of the evolution experiment (equation 77 in Levy *et al.*
2015; $90-1/s \cdot \ln[n_e \cdot s]$, where n_e is the lineage size, which is ~ 1000 cells/lineage). Thus,
mutations of such effect that occur immediately at generation 0, are nearly impossible to
be detected unless this group of mutations has a very large mutational target size or in
other words, a very high mutation rate. This matter is illustrated in detail in the
supplemental material in Levy *et al.*, Nature 2015, section 11.1, page 54.

**In our data**, even though we are not able to detect weakly adaptive mutations, such
mutations will not significantly affect the prediction of population dynamics and
evolutionary outcomes. Thus, the DFE inferred in this study provides a representative
depiction that can be applied to theoretical studies. Additionally, it is important to note
that the barcode sequencing approach used here is able to detect lineages at
frequencies as low as 0.001%, compared to the commonly used population sequencing
where only high frequency lineages (e.g. $> 1\%$) can be detected. To our knowledge, our
study has resulted in the most comprehensive understanding of DFE during the 2nd step
of adaptation. Finally, we observed similar shapes of DFE between the two steps (long
right tail and a peak at $s = \sim 4\%$) within our detection limit.

We modified our writing to incorporate the information above in lines **288-302** (Materials
and Methods, section 'DFE / Mutational Fitness Spectrum $u(s)$ inference').

*A specific class of beneficial mutation—diploidization—creates a complication for these*
*analyses, because it arises often enough to dominate the distribution of fitness effects,*
*evidently creating a distinct peak (line 589). Despite the fact that diploidization of*
*haploid yeast was observed in the preceding study of these populations (and is*
*commonly observed in yeast evolution experiments using haploids) there is no mention*
*of the issue in the introduction. This is surprising given that the appearance of diploids*

*would be an obvious prediction to make. The possible fitness benefits of diploidy are*
*interesting in their own right, and it would be great to see estimates of diploidization*
*rates and effects along with confidence intervals.”*

The introduction has been updated to cover the topic of diploidization during haploid
adaptation (lines **99-104**). Fitness effects of diploids and their 95% confidence intervals
were added for each genotype (lines **657-658**). Diploidization rates are reported in lines
**671-672**.

*The authors indicate that one of the benefits of their study is the short time scale in*
*terms of generations, but it’s not obvious why evolutionary dynamics on this timescale*
*are more relevant than longer timescales. If this were a natural population we would*
*expect continued clonal competition among adaptive genotypes, as well as*
*recombination among genotypes, which could presumably result in a single best*
*genotype on a longer timescale. In other words, evidence of historical contingency at*
*short time scales may not be sufficient to demonstrate that historical contingency is*
*important at moderately longer timescales. Indeed, even at the short timescale of the*
*first few adaptive changes studied here, the data seem to offer a fair amount of support*
*for convergent adaptive pathways rather than contingency, with mutations in two*
*particular pathways, plus diploidization, being the most common. There is therefore*
*some risk that these findings would add to, rather than clarify, the mixed evidence on*
*historical contingency described in the Introduction. It would be good to see a nuanced*
*discussion of these issues and caveats as a guide for future research.”*

We agree with the reviewer that *per se* evolutionary dynamics on shorter timescale
experiments are no more relevant than those on longer timescales. We have added a
paragraph at the discussion (2nd paragraph, lines **687-700**) explaining why our study is
important. In contrast to the reviewer’s point of view, we think that evolutionary
dynamics comparisons between a laboratory and a natural population are hard to draw.
Specifically, well-mixed environments, such as our evolutionary environment, do not
reflect any natural environment, which should be to various degrees more structured. As
a result, competition will be limited within lineage boundaries, and no single genotype is
expected to win. Finally, our results indeed support converging adaptive strategies –
see revised discussion.

*Additional comments:*

*Diploidization of the genome presumably doubles the barcode sequence, creating a*
*difference between the frequency of a barcode and the frequency individuals with the*
*corresponding genotype. Is there a need to account for this in general, and specifically*
*when estimating the fitness effects of diploidy?*

This is a good point to clarify. Diploidization of the genome does not affect fitness
estimates in both fitness remeasurements and fitness inference during evolution. In
fitness measurements, relative fitness is inferred based on the ~90% of neutral haploids
in the population by calculating the fold change of a lineage’s frequency over time.
Lineages with the same fitness have the same fold changes in terms of lineages’

frequencies. It is the change in frequency that matters for a fitness estimate rather than
the absolute frequency. During the evolution experiments, fitness is estimated based on
population mean fitness, which is calculated by using low abundant lineages as
neutrals. The fitness is then estimated by using lineages' frequency across multiple time
points. Similarly, the fold changes of lineages' frequency rather than the absolute
lineage frequency affect the fitness estimate (under the assumption that the majority of
the population has the same fitness as the founder). An instantaneous doubling of the
copy number of a barcode within a lineage happens in a single cell – i.e. 1 becomes 2
copies, not for a whole lineage simultaneously. Thus, as fitness is estimated based on
change in frequency over time, when a lineage is at an appreciable frequency, the fact
that there are two copies of the barcode within a lineage does not affect the fitness
estimate.

*The negative correlation between ancestral fitness and highest evolved fitness (lines*
*491, 586, Fig. S6) is consistent with diminishing returns, but not likely to be statistically*
*significant with just four data points. A rank correlation would be more appropriate, but*
*still won't be formally significant given the sample size. Perhaps there are more*
*powerful ways to test for this association, e.g., by making use of randomization to*
*generate an appropriate null distribution.*

Rank correlation has also been added to the legend of figure S7 (previously S6) and at
the appropriate places in the text, along with p-values and the p-values themselves are
not significant (lines 560-561 and 661). Nonetheless, such an anti-correlation is
suggestive.

*Line 600. Here and elsewhere I think it's important to clarify that this does not refer to*
*mutation rates in general, but rather the rate that mutations arose that were beneficial*
*enough to be detected.*

The writing has been modified accordingly to clarify that we refer to mutations within
some fitness interval.

*The abbreviation UMI seems to appear only twice (lines 231 and 234), and could be*
*avoided, but should otherwise be defined.*

The abbreviation was defined the first time that was mentioned (line 245-246). We have
opted to keep the abbreviated form for consistency with earlier studies.

*Reviewer #2 (Remarks to the Author):*

>> Summary

*The authors conducted a comprehensive analysis of the 2nd mutations arising from*
*three genetic backgrounds differed by one adaptive mutation (i.e. 1st mutation). They*
*found a significant difference between the overall mutation spectrums of the adapted*
*and wild-type (WT) ancestors. Among the three adapted ancestors, they found*
*mutations arising from the two Ras/PKA pre-adapted ancestors more similar than those*
*from the TOR/Sch9 pre-adapted ancestor. The authors discussed the implications of*

*their findings in historical contingency and the frequently seen diminishing trend of*
*adaptation.*

>> *Major Comments*

*Changes in the spectrum (i.e. fitness effect, probability, molecular type ...) of 1st and*
*2nd mutations in adaptation is the major observation of this work, which is both*
*interesting and important in the study of DFE. This finding could result from (1) clonal*
*interference, enriching the strongest beneficial mutation in the first step or (2)*
*antagonistic interaction between the 1st and 2nd mutations. Unfortunately, the*
*resolution of fitness assays and the analysis presented by the authors could not make a*
*clear distinction or estimation of the relative contribution between of the two causes.*
*Such limitation compromises the value and novelty of this work.*

*We agree with the reviewer that our experimental design (evolution under well-mixed*
*environment followed by clone isolation) biases towards selection of the highest fitness*
*lineages. As a result, it is unclear whether the adaptive targets during the 2nd step*
*evolutions, were not adaptive as 1st step adaptation (antagonistic interaction), or their*
*effect was sufficiently small to be outcompeted by other larger effect targets*
*(diploidization, TOR/Sch9 and Ras/PKA pathways) (clonal interference). We have now*
*clarified this point in the discussion. However, we do not think that this compromises the*
*current study, as the intended premise of the study is to compare the DFE during*
*adaptation of the different founders, rather than to detail the causes of such a shift.*

*On the other hand, while this study shows that the 2nd mutations arising from the*
*TOR/Sch9 and Ras/PKA backgrounds are somewhat different, the experimental design*
*and results do not seem suitable to address historical contingency. What the authors*
*observed in such short time frame might reflect just a transient alteration of the temporal*
*order of mutational events and may have limited impacts on the long-term evolutionary*
*trajectory. Particularly, it seems unnecessary to make a big deal about seeing*
*TOR/Sch9 mutations more likely to arise in the Ras/PKA pre-adapted ancestor, and*
*vice versa. The authors' prior work already shows the predominance of these two*
*mutation types in the first-step mutation spectrum and suggests that their beneficial*
*effects do not appear to cancel out.*

*Our study differs from previous studies in that we characterized evolution over a shorter*
*timescale and at a higher resolution, compared to previous studies that have addressed*
*the prevalence of historical contingency after hundreds of generations. To the best of*
*our knowledge, no study has been done to see how fast historical contingency and*
*diminishing returns happen (i.e. after a single mutation or after multiple mutations via a*
*long period of adaptation). We believe that our study fills the knowledge gap on whether*
*and how historical contingency and convergence manifest in the shortest timescale*
*possible, during which only one adaptive mutation is acquired.*

*In addition, we agree that it is not particularly unexpected to observe that TOR/Sch9*
*mutations are more likely to arise in the Ras/PKA pre-adapted ancestor, and vice versa.*
*It was not our intention to use the particular finding to address historical contingency.*
*Instead we considered it as an example of convergence (paragraph in lines **712-727***

specifically refers to convergence). Additionally, the fact that we observe such mutations
in all three genetic backgrounds is a good proof of concept and indicates that due to the
magnitude of their fitness effect, these mutations outcompete other adaptive mutations.

*If the authors could run the evolution experiment longer, say 1000 generations, the*
*three types of adapted ancestors might end up acquiring a similar set of mutations.*
*Such result, if occurs, would indicate the three lineages following similar evolution*
*trajectories, and as such, no historical contingency. The authors could address this*
*concern by (1) evolving their populations longer or (2) referring to the results of prior*
*evolution experiments involving similar strains and growth conditions.*

The goal of the work was to study adaptation at the smallest possible timescale (single
mutation resolution). Long-term convergence is a possibility, and as we comment, from
the mutational target it is suggestive that there is already phenotypic convergence. This
point is addressed by our current writing that covers previous comments. We agree that
longer term evolutions would be interesting and useful, though they are beyond the
scope of this work.

*Line 560-562: Results presented here contradict the message in Line 98 and Line 115.*
*The title of this section should be rephrased as well.*

The results presented in the Results section (lines 612-635) refer to an analysis with
alleles 'at hand', that is on genes that were affected in all backgrounds. As a result, our
sample should be biased towards the highest fitness alleles/ combination of alleles. The
particular analysis does not provide strong support for diminishing returns, but that does
not compromise the rest of our conclusions on diminishing returns. As a response to
this comment we added lines at the end of that particular section to avoid confusion.
The title was rephrased.

>> *Minor Comments*

*Line 37-41: The interpretation of results is a circular argument.*

We modified the abstract to eliminate this circularity.

*Line 53: Two very first studies of diminishing returns epistasis are missing here - Chou*
*(2011) and Khan (2011).*

We agree that these studies should be cited and have thus added them.

*Line 388: RAS or Ras. The style is not consistent throughout.*

All instances have been turned to 'Ras/PKA'

*Figure 2: Does "density" mean the probability density function of the mutation rate? This*
*term is not defined. Moreover, it would be **nice to show the fitness distribution of the***
***two major types of beneficial mutations** (i.e. TOR/Sch2 & Ras/PKA)*

We thank the reviewer for pointing this out. $\mu(s)*ds$ is defined as the mutation rate per
generation per cell for the mutations with fitness within the range $[s, s+ds]$. $\mu(s)$ is the
fitness-dependent probability density function of the mutation rate. The following
clarifications were added to our manuscript: Materials and Methods, section 'DFE /
Mutational Fitness Spectrum $\mu(s)$ inference', lines **271-272** (parenthesis content) and
lines **276-277**.

We agree that it would be nice to be able to generate DFE for the two major target
pathways. However, while we have genotyped thousands of evolved clones from the
evolution experiments to infer the trend of genetic basis of adaptation, the number of
genotyped adaptive lineages is not big enough to infer individual DFE spectra. The
current DFE is inferred using all barcoded lineages during evolution regardless of their
genotypes.

*Line 478-479: I am not able to see this in Fig. 3.*

Clarifications on the figure discussion were added to the paragraph starting at line **539**.

*Line 491: Report P value here.*

p values were added (lines **560** and **661** and figure S7 legend)

*Line 535: Table or table? Style consistency.*

All occurrences have been turned into 'Table'

*Line 611-613: Recent studies of global fitness landscapes, such as Diss (2018),*
*Poelwijk (2019), Bendixsen (2019), and Kuo (2020), are worth comparing here.*

We have added some text to the second paragraph of the discussion citing these
references.

*Line 617: Correct font style.*

Font style (italic) was there for emphasis. However, it has been turned to non-italic.

*Line 631-632: Discussion of sign epistasis appears out of blue.*

The particular study (Ono *et al.*, Plos Biol 2017) is mentioned because they observed
sign epistasis among adaptive variants in the same pathway. Here, in most cases we
did not recover second step adaptive variants affecting the same pathway as the first
step, which can presumably be caused by sign epistasis among mutations in the same
pathway. The previous study suggests that if we start combining mutations (at least first
step mutations) that affect the same pathway, we may frequently observe sign epistasis
between them. We have modified our writing (lines **721-727**) to better provide such a
connection.

*Line 665-666: Add "in the subsequent step."*

'during subsequent adaptation steps' was added at the end of the sentence (line 778)

Reviewer #3 (Remarks to the Author):

*In this paper Aggeli et al., study the distribution of fitness effects during the second step*
*of adaptive evolution in an experimental evolution regime. Starting with three strains*
*that were recovered from a prior evolution experiment containing adaptive mutations in*
*TOR1, CYR1 and GPB2 the authors introduced a complex library of random molecular*
*barcodes into each strain. They then performed experimental evolution using serial*
*batch dilution. Using the dynamics of the barcoded lineages they estimate the*
*distribution of fitness effects for mutations acquired during the second step of adaptive*
*evolution. Using whole genome sequencing of lineages they identify the molecular basis*
*of the second adaptive mutation. They conclude that **their results are consistent with***
***diminishing return epistasis.***

*This is an interesting and well performed study. Although the conclusion is consistent*
*with prior studies, it provides additional evidence for the **generalizability of global***
***epistasis in adaptive evolution**, which is of importance to the field. Moreover, there*
*are several **interesting observations regarding the functional relationship between***
***targets of adaptive mutations following the acquisition of a beneficial mutation***
***that are of interest to the field.** Prior to publication, the authors should consider the*
*following points:*

*-In the introduction the authors state that they have characterized the rate of adaptation,*
*but I do not find this in the paper. I would expect to see a quantification of the rate of*
*increase in population fitness. Is this possible to infer with the data? This would be an*
*interesting result to understand how initial population fitness impacts the rate of*
*adaptation*

*This information is provided in Table 2, which includes the mutation rate for*
*diploidization, mutations with fitness >0.05, mutations with fitness between 0.07 and*
*0.12 and mutations with fitness > 0.12. As a response to this comment, we added a*
*column to this table, that includes the total mutation rate with fitness within our detection*
*limit (>0.02 fitness coefficient with respect to the respective ancestor). We included the*
*relevant commentary in our manuscript (lines 508-513).*

*-There is no data showing the benomyl assay for detecting diploids. I think that some*
*representative result should be presented - even if it has been shown in a prior*
*publication - and an explanation as to why a more sensitive assay such as DNA staining*
*and FACS wasn't used.*

*A representative pair of plates was added (control/ +benomyl) as figure S4A. The*
*benomyl-based ploidy assay used here has been developed and validated against DNA*
*staining with Propidium Iodide and flow cytometry as part of Venkataram et al., Cell*
*2016 (see STAR Methods). The reason why benomyl assay was chosen is because it*

can be easily scaled up to assay thousands of lineages, which is more time and
resource efficient than DNA staining and FACS.

*-The initial beneficial mutations are referred to as “presumptive gain of function”, but*
*there is no evidence that this is the case and this terminology implies that the activity of*
*these pathways is increased, which has not been shown. Why not just call them*
*beneficial mutations? Or demonstrate that they are indeed gain of function using*
*reporter, or phosphorylation, assays. Moreover, two of the studied mutations are not*
*loss of function mutations which conflicts with the authors’ statements about the*
*prevalence of loss of function mutations in the first adaptive step.*

Presumptive gain of function mutations were inferred mainly by the regulatory role that
the affected proteins have on the signaling pathway (see Fig 3 on Venkataram *et al.*,
Cell 2016). It is expected that adaptation affects a given pathway in a consistent way, so
for example if selection favors LOF mutations in negative regulators, it should also favor
GOF mutations in positive regulators. This assumption is consistent with the type of
mutations observed (GOF should be missense mutations only, whereas LOF can be
any kind of mutation, and are frequently frameshift or nonsense mutations); additionally,
GOF mutations are rarer than LOF mutations. In our prior paper, we observed many
LOF mutations in negative regulators in the Ras/PKA pathway, but only rarely saw
mutations in the positive regulators, and they were always missense mutations. We
have added text to the manuscript to clarify this (lines 436-448).

For our experiments, we used one mutant representing each of the following categories:
a GOF and a LOF mutant affecting the Ras/PKA pathway and a GOF mutant affecting
the TOR/Sch9 pathway. No LOF mutant affecting the TOR/Sch9 pathway was available.
We intentionally chose the particular mutants to diversify the backgrounds explored
(lines 448-450).

*Figure 2 reproduces the same wildtype data three times. Does plotting the three*
*different distributions of second step mutations on the same plot make it possible to*
*compare between the different backgrounds? Perhaps it would be too cluttered, but it*
*would be useful to see the extent to which the DFE differs between the three*
*backgrounds. The error bars may be unnecessary for this visualization.*

We agree that it will be too cluttered to put all DFE on the same plot. However, we
added a supplementary figure with all DFE together and without the error bars to help
readers who are interested in comparing them (Figure S6).

*-Figure S7 needs a quantification of the correlation*

Pearson correlations and p-value were added on the figure legend (this is Figure S8
now).

Reviewers' Comments:

Reviewer #1:

Remarks to the Author:

While the authors were able to clarify certain aspects of their work, certain limitations of this approach remain, with potential affect the generality of these results.

While the authors generally treat diploidization as simply another adaptive pathway, it's not clear that they have fully worked through the implications that this unique class of mutation will have for their approach. Given the high mutation rate from haploidy to diploidy there will be thousands of diploid cells produced each generation in the large experimental populations the authors studied. As a consequence, the overall frequency of diploids in the population will go up steadily due to mutation pressure even if they are selectively neutral. These diploids will presumably occur within all barcode groups, leading to clonal competition among many strains with the same mutation (diploidy). If diploidy is truly beneficial its fitness effect in the evolution experiments would likely be underestimated due to clonal competition and the misclassification of some lines as neutral. Another possible outcome of a high mutation rate to diploidy is that weakly-beneficial or neutral ploidy variants will be more likely to overcome drift and reach high frequency, relative to point mutations with similar fitness effects but lower mutation rates. The authors have made an effort to address issues surrounding diploidy, but it's hard to ignore the potential for diploidization to skew the inferred DFE, both by generating a distinct peak and by dominating competition among lineages.

Given the authors' response to Reviewer 2 "the intended premise of the study is to compare the DFE during adaptation of the different founders, rather than to detail the causes of such a shift" (line 183), I would suggest revising the section of the manuscript that claims the study provides "substantial insight into the potential causes underlying diminishing returns and historical contingency" (line 89).

Regarding alleles with $s < 2\%$, line 498-9 states "such lineages do not significantly affect the prediction of population dynamics and evolutionary outcomes (see Material and Methods)," and this sentiment is repeated in the authors' rebuttal. I assume this refers to the evolutionary outcomes on this particular experimental timescale, in a well-mixed population, in this particular environmental and genetic context; in general such mutations are probably highly important to adaptation, as implied in the Discussion (line 695). The authors should clarify these statements to avoid misleading a general audience. Rather than claiming that such alleles are irrelevant, it seems to me more appropriate to say that this version of the DFE is truncated due to experimental limitations. We'd expect many alleles with $s > 2\%$ that appear in the DFE results to be ultimately outcompeted by more-beneficial alleles, making them effectively irrelevant as well. If we treat the range of possible beneficial mutations as a distribution (rather than seeking to identify a single best mutation), then all alleles with $s > 0$ would seem to be relevant. While undoubtedly more detailed than previous efforts, describing the observed DFE as comprehensive seems to run counter to the stated limitations of the approach.

Reviewer #2:

Remarks to the Author:

The aims and strength of this work are better explained in the revised manuscript. Yet it is still doubtful to link one-step mutational events with historical contingency, which is defined as chance-influenced events with substantial "long-term" effects. Moreover, two newly added paragraphs in Discussion require clarification:

1. The second paragraph in Discussion (Line 687-700) is confusing and incoherent with the results and aims of this work. Do the results of this work demonstrate the fitness landscape as smooth or rugged? If so, please elaborate. In addition, "peaks" do not fit the context of Line 689-691 well. Peaks imply that genotypes are trapped by fitness valleys and require more than one mutational event in order to

escape and adapt. Finally, comparing this study (short-term evolution in well-mixed environment) to the evolutionary outcomes in structured environments in Nature appears unnecessary and out of the context.

2. Convergent evolution refers to the process whereby organisms "not" closely related, independently evolve similar traits as a result of having to adapt to similar environments or ecological niches. Since the wild-type is the ancestor of the adaptive ancestors and differs by just one mutation in this study, the comparison described in Line 712-727, particularly Line 714-718, is inappropriate and not meaningful.

Reviewer #3:

Remarks to the Author:

This resubmission has adequately addressed most of my comments on the initial submission. However, there are some points that I think the authors should address prior to publication.

I don't believe that the first point made in my initial review has been addressed. The authors have determined the rate at which new beneficial mutations are acquired and the fitness effects of those mutations. Given the quantification of fitness effects and the frequency of those lineages it should be possible to compute the mean population fitness. Quantifying the rate with which mean population fitness changes in these adapted lineages would be interesting and support the claim of diminishing returns epistasis.

Values in Table 2 should contain units for each metric. I would also suggest adding additional information to the Table heading as it is hard to understand the values in this table without reference to the text.

The y axis in figure 2 is confusing. I understand these distributions to be probability densities, but I do not understand what the $u(s)$ represents. This is true of other plots in the paper.

Figure 3 y-axis should contain units. This is also true of other plots throughout the paper.

Response to Reviewer Comments

We thank the reviewers for their comments – below we provide a point-by-point response to each of the comments. Our responses are highlighted, while the reviewers' comments are italicized.

REVIEWER COMMENTS¶

Reviewer #1 (Remarks to the Author):

While the authors were able to clarify certain aspects of their work, certain limitations of this approach remain, with potential affect the generality of these results.

While the authors generally treat diploidization as simply another adaptive pathway, it's not clear that they have fully worked through the implications that this unique class of mutation will have for their approach. Given the high mutation rate from haploidy to diploidy there will be thousands of diploid cells produced each generation in the large experimental populations the authors studied. As a consequence, the overall frequency of diploids in the population will go up steadily due to mutation pressure even if they are selectively neutral. These diploids will presumably occur within all barcode groups, leading to clonal competition among many strains with the same mutation (diploidy). If diploidy is truly beneficial its fitness effect in the evolution experiments would likely be underestimated due to clonal competition and the misclassification of some lines as neutral. Another possible outcome of a high mutation rate to diploidy is that weakly-beneficial or neutral ploidy variants will be more likely to overcome drift and reach high frequency, relative to point mutations with similar fitness effects but lower mutation rates. The authors have made an effort to address issues surrounding diploidy, but it's hard to ignore the potential for diploidization to skew the inferred DFE, both by generating a distinct peak and by dominating competition among lineages.

We agree that diploids (like other types of mutation) will keep feeding into the population. However, the constant feeding of diploids does not necessarily lead to their increased frequency for the following two reasons:

Firstly, the frequency dynamics of diploids are determined not only by how rapidly diploids enter the population, but also by how they fluctuate once they arise. Variation in the offspring number of diploids introduces stochasticity, which can lead to diploids' extinction (drift out). If a diploid does not go extinct, its population size will eventually be large enough that it then grows exponentially and essentially deterministically: i.e. it "establishes". The number of diploids needed to avoid stochastic fluctuations is inversely proportional to the fitness value of diploids relative to the population mean fitness: $n(est) \sim 1/s$, where $n(est)$ is the number of diploids required to reach establishment size. Thus, the higher a mutant's fitness is, the smaller the $n(est)$ is.

Secondly and most importantly, the population mean fitness increases over time in our large and well-mixed populations; consequently, the fitness of diploids relative to the population mean fitness constantly decreases as evolution proceeds.

Below is a back-of-the-envelope calculation to support our argument taking the above two reasons into account:

With an effective population size of $\sim 6E8$ and an estimated diploidization rate of $\sim 1E-5$ per cell per generation in the wild-type background (similar to the estimates in Harari et al., 2018), we expect that **$\sim 6,000$ ($6E8 \times 1E-5$) diploids constantly feed into the population during each generation.**

At time 0, population mean fitness is close to 0. The fitness of diploids (s) relative to the WT ancestor is ~ 0.04 per generation by our measurements. Given $n(est) \sim 1/s$, **diploids will get established in the population as long as >25 diploids ($=1/0.04$) enter the population**, a number much lower than the 6,000. Thus, at time $0+1$, diploids established and increase their frequency at a rate of $e^{(s \cdot t)}$.

Over time, other beneficial mutations will also establish and rise in frequency, and together with the diploids will increase the population mean fitness. **Due to this increased population mean fitness, the relative fitness of diploids decreases and newly arising diploids are less and less likely to become established. Once the population mean fitness reaches ~ 0.04 (relative to WT ancestor), the effective relative fitness of diploids becomes 0 (or neutral), meaning that newly arising diploids cease to establish in the population, despite their high mutation rate (the constant feeding).**

Consistent with the above explanation, we observed that in our wild-type evolving population, the population mean fitness approaches 0.04 between 90-100 generations, which coincided with a decline in the overall frequency of diploids, indicating that new diploids no longer establish (Figure S4B, and Figure S7 in Venkataram et al); note, diploids do subsequently further increase in frequency after that dip, but due to second beneficial mutations occurring on a diploid background. Another point of note, is that the above overestimates the likelihood of establishment and observation of diploids – newly arising beneficial mutations also have an establishment time (the time they take to reach establishment size from when they arose), which further limits their likelihood of observation, because by the time they reach establishment, the population mean fitness may have already passed them by (see Figure 3 in Levy et al).

With regards to *"If diploidy is truly beneficial its fitness effect in the evolution experiments would likely be underestimated due to clonal competition and the misclassification of some lines as neutral"*. We agree that clonal interference plays an important role in our evolution experiments, as would be expected in a well-mixed, large population with increasing mean fitness. The fitness estimates from our evolution experiments should not be greatly affected when adaptive lineages are at a low frequency. As a result, early emerging diploids will be identified as adaptive mutants. As population mean fitness increases, the effect of clonal interference becomes more significant; newly emerged diploids are effectively less beneficial and less likely to be established in the population, making it difficult to accurately estimate diploids' fitness at this point. **Fortunately, we are able to assess to what extent clonal interference**

affects the fitness estimates. By isolating thousands of lineages from the evolving population and competing them with wild-type ancestor, we are able to precisely (re-)measure the fitness of these evolved lineages, which is referred to as fitness remeasurements in the manuscript. We then use the fitness remeasurements as a gold standard to evaluate how well we do in fitness estimates using our evolution data. We find that **the fitness remeasurements are largely consistent with the fitness estimates from the evolution data (see Figure S8), which is also consistent with our prior experience (see Figure 2D in Levy et al, and Figures 2E and F in Venkataram et al).** Thus, we conclude that our DFE is not (or at least not obviously) skewed by the high diploidization rate.

*Given the authors' response to Reviewer 2 "the intended premise of the study is to compare the DFE during adaptation of the different founders, rather than to detail the causes of such a shift" (line 183), I would suggest **revising the section of the manuscript that claims the study provides "substantial insight into the potential causes underlying diminishing returns and historical contingency" (line 89).***

We agree with the reviewer, and have revised this sentence to instead read "*This allowed us to deeply characterize the beneficial mutational spectra and fitness effects, and draw direct comparisons among second-step and between first and second-step adaptations*"

Regarding alleles with $s < 2\%$, line 498-9 states "such lineages do not significantly affect the prediction of population dynamics and evolutionary outcomes (see Material and Methods)," and this sentiment is repeated in the authors' rebuttal. I assume this refers to the evolutionary outcomes on this particular experimental timescale, in a well-mixed population, in this particular environmental and genetic context; in general such mutations are probably highly important to adaptation, as implied in the Discussion (line 695). The authors should clarify these statements to avoid misleading a general audience. Rather than claiming that such alleles are irrelevant, it seems to me more appropriate to say that this version of the DFE is truncated due to experimental limitations. We'd expect many alleles with $s > 2\%$ that appear in the DFE results to be ultimately outcompeted by more-beneficial alleles, making them effectively irrelevant as well. If we treat the range of possible beneficial mutations as a distribution (rather than seeking to identify a single best mutation), then all alleles with $s > 0$ would seem to be relevant. While undoubtedly more detailed than previous efforts, describing the observed DFE as comprehensive seems to run counter to the stated limitations of the approach.

We agree that the DFE (by definition!) is a distribution, but still contend that at least in our experiments, that mutations of $s < 2\%$ will not greatly affect the dynamics or the outcome of our evolution experiments – if we had a more fit genotype, or a different experimental condition, where a 2% fitness gain was the largest possible adaptive mutation, of course they would matter. However, in our experimental condition, where there are mutations with effects $> 5\%$, they do not – from the discussion of Venkataram et al:

“Note, we have not attempted to identify every potentially adaptive mutation in our experimental condition, rather we have identified most of the mutations that drive or are likely to drive the evolutionary dynamics of our system. In this system, with its well-mixed population, any adaptive mutation that is either too selectively weak or has a very low rate of occurrence cannot effectively drive the adaptive dynamics, because of clonal interference (Levy et al., 2015). For example, if the target sizes for adaptive mutations in two genes are k_1 and k_2 respectively, with selective advantages s_1 and s_2 , then after a time T in a large population the ratio of the fractions of the population of the two classes of mutants are $k_1 \exp(s_1 T)$ and $k_2 \exp(s_2 T)$. If $T = 88$ generations, as for our sampled clones, with $s_1 - s_2 = 5\%$, and the same target sizes ($k_1 = k_2$), the mutant with 5% greater fitness benefit will be observed 100 times as often. However, the mutational target size is also important: if k_2 were 100x larger than k_1 (e.g., k_2 includes many possible beneficial loss of function mutations while k_1 includes only very few beneficial gain of function mutations), this compensates for the selective effect and mutations in the two genes will become comparable fractions of the population. Therefore, both selective advantage and the mutational target size are important in determining which mutations drive adaptive evolution.”

While for diploids, this reasoning explains why we see so many (they are less fit than the fittest mutants, but arise at a higher rate), for mutations with a 2% fitness benefit that arise at a similar mutation rate as those with a 7% fitness benefit, they would be unlikely to be seen, and readily surpassed by the population mean fitness.

However, the following modification (in italics) was included in the results: *‘While lineages with such small effect mutations might contribute to evolutionary outcomes of populations under weaker selection pressures (for example smaller bottlenecks or structured environments), they do not significantly affect such outcomes in our experiments, which have large population sizes and are not mutation limited (see Material and Methods)’* The previous sentence was also modified to *‘Note that the DFES presented are truncated for lineages with $s < 0.02$, as this fitness coefficient approaches our lineage tracking data detection limit.’* Additionally, the penultimate paragraph in the discussion now elaborates on that limitation of our system.

Reviewer #2 (Remarks to the Author):

The aims and strength of this work are better explained in the revised manuscript. Yet it is still doubtful to link one-step mutational events with historical contingency, which is defined as chance-influenced events with substantial “long-term” effects. Moreover, two newly added paragraphs in Discussion require clarification:

We think our usage of “historical contingency” is consistent with that in the experimental evolution literature (e.g. Blount et al), where “historical” has not necessarily implied a specific timescale – however, if the reviewer believes that it generally implies a long time scale, then we would prefer to not confuse readers; hence we have modified the language in the abstract and the discussion to refer to contingency, rather than historical contingency.

1. *The second paragraph in Discussion (Line 687-700) is confusing and incoherent with the results and aims of this work. Do the results of this work demonstrate the fitness landscape as smooth or rugged? If so, please elaborate. In addition, "peaks" do not fit the context of Line 689-691 well. Peaks imply that genotypes are trapped by fitness valleys and require more than one mutational event in order to escape and adapt. Finally, comparing this study (short-term evolution in well-mixed environment) to the evolutionary outcomes in structured environments in Nature appears unnecessary and out of the context.*

The second paragraph in the discussion was added mainly as part of our response to reviewer's comments in the 1st round of revision, including the comment "Line 611-613: Recent studies of global fitness landscapes, such as Diss (2018), Poelwijk (2019), Bendixsen (2019), and Kuo (2020), are worth comparing here". Briefly, we thought that our system is quite different from the systems described in the above citation to allow for a fair comparison (see response in 1st round), nevertheless the references were included as part of this second paragraph. At second glance, we agree that the paragraph does not fit into the scope of this study, so it has been eliminated (along with the references to the above citations). Instead, a paragraph elaborating on the main limitations of our system was added (see paragraph before last).

2. *Convergent evolution refers to the process whereby organisms "not" closely related, independently evolve similar traits as a result of having to adapt to similar environments or ecological niches. Since the wild-type is the ancestor of the adaptive ancestors and differs by just one mutation in this study, the comparison described in Line 712-727, particularly Line 714-718, is inappropriate and not meaningful.*

Similar to historical contingency, convergence has sometimes been used in the experimental evolution field in a different sense – for example, Lenski (PMID 28731830) says "I often refer broadly to divergence and convergence, with the understanding that convergence may, depending on context, include some changes that are, in a strict sense, parallel." As above, however, we seek for our manuscript to appeal to as broad an audience as possible, and so as to not confuse the issue, the paragraph was modified and claims of convergent evolution were eliminated.

Reviewer #3 (Remarks to the Author):

This resubmission has adequately addressed most of my comments on the initial submission. However, there are some points that I think the authors should address prior to publication. I don't believe that the first point made in my initial review has been addressed. The authors have determined the rate at which new beneficial mutations are acquired and the fitness effects of those mutations. Given the quantification of fitness effects and the frequency of those lineages it should be possible to compute the mean population fitness. Quantifying the rate with which mean population fitness changes in these adapted lineages would be interesting and support the claim of diminishing returns epistasis.

We have calculated the population mean fitness using the same code that we used in Levy et al (see inset in Figure 2 of that paper). Note, the population mean fitness estimate is quite noisy due to the lower sequencing coverage in this study compared to Levy et al. Population mean fitness at a given timepoint T is calculated using the data at timepoint T and its subsequent timepoint and the fact that only two data points are used to estimate population mean fitness at a certain time results in high estimation noise. By contrast, lineages' fitness during the evolution were estimated using data from all timepoints and are thus more tolerant to noise.

Values in Table 2 should contain units for each metric. I would also suggest adding additional information to the Table heading as it is hard to understand the values in this table without reference to the text.

Table 2 was modified. Also, additional information was added with respect to the source of the data, and the units for mutation rate and fitness on the table heading.

The y axis in figure 2 is confusing. I understand these distributions to be probability densities, but I do not understand what the $u(s)$ represents. This is true of other plots in the paper.

Indeed, it is difficult to explain this abstract unit (probability density). To improve the clarity, we have added an example to illustrate how to calculate the mutation rate of mutants with a fitness >0.03 and <0.05 per generation (see current figure 2). Additionally, we added the following sentence “*The integration of the area below the probability density curve represents mutation rate at the respective fitness interval. For instance, the shaded area in the top panel represents the mutation rate for mutants with a fitness value >0.03 and <0.05 .*” to figure 2 legend.

Figure 3 y-axis should contain units. This is also true of other plots throughout the paper.

Fitness coefficient (here and in prior literature) is the slope of $\text{nat.log}(\text{evolved}/\text{neutral})$ over time and is usually annotated with (s). We added (s) in all fitness axes (see below) that fitness is expressed per generation and fixed formatting inconsistencies.

Fitness appears in/ we label as/ changed it to:

Fig 1 – y axis/ ‘Fitness per cycle’/ (left this one as is)

Fig 2, S5 (B & C) and S6 – x axis/ ‘Fitness per Generation (s)’/ ‘Fitness per generation (s)’

Fig 3 – y axis/ ‘Relative Fitness’/ ‘Fitness per generation (s)’

Fig 5 – y axis/ ‘Relative fitness’/ ‘Fitness per generation (s)’

Fig S1 – y axis/ ‘relative fitness’/ ‘Fitness per generation (s)’

Fig S5 (A) – x and y axes/ 'Fitness from ... dataset'/ 'Fitness per generation (s) from ... dataset'

Fig S7 – x axis/ 'Ancestor fitness'/ 'Ancestor fitness per generation (s) relative to wild-type ancestor' and y axis/ 'Fitness gain'/ 'Evolved fitness per generation (s) relative to corresponding ancestor'

Fig S8 – x axis/ 'Fitness Re-Measurement'/ 'Fitness per generation (s) from re-measurements' and y axis/ 'Evolution Fitness Inference'/ 'Fitness per generation (s) from evolutions'

Reviewers' Comments:

Reviewer #1:

Remarks to the Author:

In their responses the authors have outlined some thoughtful arguments and caveats of their study that would be useful for interpreting and expanding on their work in the future, particularly regarding diploidization. These should be incorporated into the paper as much as possible, at least as supplementary material, or perhaps by publishing the reviews. While it may be the case that the high rate of diploidization does not bias measurement of a DFE for other mutations, as the authors assert, readers should be given a chance to evaluate these arguments themselves. More generally, if the authors had the opportunity to start this series of experiments again, would they still use haploids? My view is that ploidy is often intertwined with questions about budding yeast evolution, and so it would be good to address this important issue more directly.

I have no further comments on this manuscript.

REVIEWERS' COMMENTS

Reviewer #1 (Remarks to the Author):

In their responses the authors have outlined some thoughtful arguments and caveats of their study that would be useful for interpreting and expanding on their work in the future, particularly regarding diploidization. These should be incorporated into the paper as much as possible, at least as supplementary material, or perhaps by publishing the reviews. While it may be the case that the high rate of diploidization does not bias measurement of a DFE for other mutations, as the authors assert, readers should be given a chance to evaluate these arguments themselves. More generally, if the authors had the opportunity to start this series of experiments again, would they still use haploids? My view is that ploidy is often intertwined with questions about budding yeast evolution, and so it would be good to address this important issue more directly.

I have no further comments on this manuscript.

Response: We thank reviewer for the insightful comments. We have added our responses to the Supplemental Information and the discussion in the main text. As to reviewer's question whether we would still use haploids if we had the chance to re-do the experiments, we don't have a definitive answer. Starting the evolution with haploids has its advantages and disadvantages. Without prior information, we can see merits of starting with both haploids and diploids in order to answer the questions we are interested in. As the reviewer pointed out, ploidy is an integral part of the budding yeast evolution and by starting with haploids, our experiments may not reflect what happens under natural conditions. However, it is appealing to use haploids due to the ease of identifying causative mutations that drive the evolution, as well as to compare to prior data. In the future, it will be interesting to do such experiments in diploids and see how ploidy affects further adaptation.